# Bacteriophage DNA induces an interrupted immune response during phage therapy in a chicken model

Magdalena Podlacha [1,5], Lidia Gaffke [1,5], Łukasz Grabowski[1], Jagoda Mantej [2], Michał Grabski[1], Małgorzata Pierzchalska [3], Karolina Pierzynowska[1], Grzegorz Węgrzyn [1] ✉ & Alicja Węgrzyn [4] ✉

One of the hopes for overcoming the antibiotic resistance crisis is the use of bacteriophages to combat bacterial infections, the so-called phage therapy. This therapeutic approach is generally believed to be safe for humans and animals as phages should infect only prokaryotic cells. Nevertheless, recent studies suggested that bacteriophages might be recognized by eukaryotic cells, inducing specific cellular responses. Here we show that in chickens infected with *Salmonella enterica* and treated with a phage cocktail, bacteriophages are initially recognized by animal cells as viruses, however, the cGAS-STING pathway (one of two major pathways of the innate antiviral response) is blocked at the stage of the IRF3 transcription factor phosphorylation. This inhibition is due to the inability of RNA polymerase III to recognize phage DNA and to produce dsRNA molecules which are necessary to stimulate a large protein complex indispensable for IRF3 phosphorylation, indicating the mechanism of the antiviral response impairment.

Bacteriophage therapy, or phage therapy, is the method of treatment of bacterial infections by using bacteriophages (or shortly phages), obligatory viral parasites which multiply in bacterial cells[1,2]. It is considered a potential therapy for human and animal diseases caused by pathogenic bacteria[3]. Particularly, it may be taken into consideration in the era of the antibiotic crisis which is caused by the appearance of bacterial strains resistant to many, if not all, currently available antibiotics[4]. Nevertheless, despite many years of studies on phage therapy, this method remains largely at the level of laboratory studies, experimental therapy or early stages of clinical trials[5–7]. Controversy still exists regarding its efficacy and safety, particularly when used at a large scale in medicine and veterinary medicine[8–16].

Although bacteriophages can only actively infect prokaryotic cells and are therefore considered totally safe for humans and animals[17], the results of recent studies indicated that these viruses can be recognized by eukaryotic cells (including those of humans and animals), and

induce specific reactions[10]. Bacteriophages were even suggested as human pathogens, due to both changing gut microbiome, modulating immune responses, and interacting with eukaryotic cells[18,19]. Other evidence suggests that phages are not neutral for human or animal hosts[10]. In fact, these viruses are important components of the human gut microbiome, and they highly predominate in the human virome, being named "phageome"[20]. Contrarily, if one considers phage therapy, bacteriophages should be applied in huge amounts ($10^9$ virions or higher) to ensure some efficacy, as a low dose of phages, like $10^2$ virions, was clearly demonstrated (although accidentally and by error, but providing clear evidence) to be completely ineffective during a clinical trial[21]. Therefore, although the immune system is 'familiar' with phages, it responds to a huge number of these viruses introduced suddenly into the human or animal organism. Because of the huge diversity of bacteriophages[22–24], including various types of their genetic material (dsDNA, ssDNA, ssRNA, or dsRNA), human and animal

[1]Department of Molecular Biology, Faculty of Biology, University of Gdansk, Wita Stwosza 59, 80-308 Gdansk, Poland. [2]Univentum Labs, Bażyńskiego 4, 80-309 Gdansk, Poland. [3]Department of Biotechnology and General Technology of Foods, Faculty of Food Technology, University of Agriculture, Balicka 122, 30-149 Cracow, Poland. [4]Phage Therapy Center, University Center for Applied and Interdiciplinary Research, University of Gdansk, Kładki 24, 80-822 Gdansk, Poland. [5]These authors contributed equally: Magdalena Podlacha, Lidia Gaffke. ✉e-mail: grzegorz.wegrzyn@ug.edu.pl; alicja.wegrzyn@ug.edu.pl

cells may respond differentially to different phages. For example, RNA genome phages might be recognized by the TLR3 receptor while those bearing double-stranded DNA genomes may be detected through the TLR9 receptor[25]. In both cases, interferon molecules are produced, nevertheless, details of the molecular signaling and specific pathways involved in this process are largely unknown[26–28].

The aim of this study was to identify the molecular mechanism of phage DNA recognition and the response of eukaryotic hosts. As a model, we have used an experimental phage therapy of chickens infected with *Salmonella enterica*. Our previous studies have shown that such treatment is efficient in eradicating pathogenic bacteria[29]. Analyses of chicken gut microbiomes demonstrated that treatment with the phage cocktail caused only temporary changes in the composition of the microbiota which normalized within a couple of weeks, with pre-dominance of *Lactobacillaceae* and *Enterobacteriaceae*. This was in contrast to birds treated with antibiotics (enrofloxacin or colistin) where the microbiomes remained significantly changed until the end of the experiment[29]. However, we have also demonstrated that orally administered bacteriophages could be found in various chicken organs, including the gut, liver, spleen, muscle, heart, kidney, and brain[29]. This raises the question about the safety of phage therapy. Interestingly, subsequent work demonstrated that the immune homeostasis of chickens was not disturbed after administration of a concentrated phage cocktail (the dose of $10^9$ plaque forming units (PFU) of each of two phages), as indicated by a lack of cytokine imbalance, unchanged ratios of key immune cell subpopulations, and a lack of the stress axis hyper-activity (features that were detected as adverse effects of antibiotic therapy, performed in the same set of experiments)[30]. Indeed, increased levels of anti-inflammatory cytokines (IL10 and IL4) were observed in phage-treated chickens[30]. This raised the question about the molecular processes that allow bacteriophages to be recognized as viruses by human or animal cells, but also lead to anti-inflammatory (rather than pro-inflammatory) reactions. Therefore, here we aimed to solve this problem by determining molecular pathways of the signals which are induced by the presence of orally administered bacteriophages in the blood of chickens, while leading to anti-inflammatory effects.

## Results

### Bacteriophage DNA is present for at least 5 weeks in the blood of chickens during phage therapy, causing elevation of IFNβ levels

In the experimental system employed in this study, chickens ($n = 100$) were divided into four groups, receiving (1) 0.89% NaCl (negative control), (2) bacteriophage cocktail (composed of a mixture of two bacteriophages with dsDNA genomes, capable of infecting *S. enterica* serovar Typhimurium, vB_SenM-2 and vB_Sen-TO17, administered at the dose of $10^9$ PFU per an animal each), (3) *S. enterica* serovar Typhimurium (at the dose of $10^6$ colony forming units (CFU) per an animal), (4) *S. enterica* serovar Typhimurium and the phage cocktail (at doses indicated above). *S. enterica* was administered orally on the first day of the experiment, while the phage cocktail was administered orally, daily for 2 weeks, starting from the second day (Fig. 1a). Samples of blood were collected on days 6, 20, 28, and 34 after five chickens from each group were sacrificed.

The presence of phage DNA in blood samples was detected by PCR (the detection limits in the PCR analyses were determined to be 100 fg and 10 fg for vB_SenM-2 and vB_Sen-TO17 phage DNAs, respectively (corresponding to $6.19 \times 10^2$ and $2.08 \times 10^2$ molecules of genomic DNA of vB_SenM-2 and vB_Sen-TO17, respectively); Supplementary Fig. S1). Interestingly, the genetic material of both bacteriophages administered in the cocktail was detected in all samples in groups where phages were given, during the whole experiment (Fig. 1b, c). Investigating the major factors produced in response to infections, we have tested levels of interferons α and β (IFNα and IFNβ) and TNFα (note that the level of IFNγ was demonstrated previously to be significantly increased in chickens infected with *S. enterica* but not

in those treated with the phage cocktail alone or with both *S. enterica* and the phage cocktail)[30]. Levels of both IFNα and TNFα were strongly elevated (relative to the group of control chickens, treated solely with 0.89% NaCl) in birds infected with *S. enterica* but not in the group treated with the phage cocktail. In fact, treatment with both bacteria and phages reduced the levels of IFNα and TNFα relative to the *S. enterica*-treated group, but not normalized the abundance of these factors to the level characteristic of the control group (Fig. 1d, f). Conversely, the levels of IFNβ were increased at all tested time-points also in blood samples of chickens treated solely with bacteriophages (Fig. 1e). Therefore, these results indicated that blood cells responded to the treatment with the high dose of phages by producing IFNβ.

### cGAS- and TLR9-dependent processes in the recognition of bacteriophage DNA by chicken cells

There are two major pathways of viral DNA recognition which lead to the production of IFNβ by animal/human cells, dependent on cGAS or TLR9. Therefore, we have tested the levels of major components of these pathways in the employed experimental system. We found that although levels of cGAS (the cytosolic sensor of viral DNA), IFI16, and DDX41 were elevated in blood cells of chickens infected by *S. enterica*, they were comparable to controls in the birds treated with the phage cocktail (Fig. 2b and Supplementary Fig. S2). When both *S. enterica* and the phages were administered, the levels of these factors were significantly lower relative to *S. enterica*-infected animals (Fig. 2b and Supplementary Fig. S2).

However, levels of TLR9 were considerably increased not only in chickens infected with *S. enterica* but also when the phage cocktail was administered without bacterial infection (Fig. 2a). This strongly suggested that phage DNA was recognized by the TLR9 receptor. Such a possibility was corroborated by finding that in the presence of both *S. enterica* and the phage cocktail, the levels of TLR9 were even higher than when chickens were treated solely with either the bacteria or the phages (Fig. 2a). As expected, levels of other tested TLR receptors (TLR3, TLR4, and TLR6) were not elevated in phage cocktail-treated birds, but infection with *S. enterica* resulted in their significantly higher abundance which was reduced or even corrected by the simultaneous presence of bacteriophages (Supplementary Fig. S3).

In the absence of elevated levels of both cGAS and STING (TMEM173) in the blood samples of chickens treated solely with the phage cocktail, the abundance of cGAMP (a specific signal nucleotide) was significantly increased under these conditions (Fig. 2b–d), indicating that this pathway was also induced. Nonetheless, it should be noted that not the specific levels of cGAS and STING, but their specific activities are crucial in the signal transduction process. This was confirmed by the especially high levels of IRF3, the final component of this signal transduction pathway (Fig. 2e). However, IRF3 (a transcription factor) is not able to enter the nucleus and activate transcription without phosphorylation, and levels of the phosphorylated form of this protein were very low in birds treated solely with the phage cocktail (Fig. 2f). This indicated that the signal transduction pathway which uses cGAMP as a mediator is induced in blood cells of birds treated with bacteriophages, however, it is stopped at the stage of IRF3 phosphorylation. The specific inhibition of the signal transduction pathway at the stage of the IRF3 phosphorylation process was corroborated by demonstrating the elevated levels of proteins (TANK, TBKBP1, and NAP1) involved in the complex activating this modification of IRF3 (IKKε, TBK1 and its phosphorylated form) (Fig. 3a–f).

Contrary to phages, the presence of *S. enterica* caused an increase in levels of cGAMP, STING, IRF3, and phospho-IRF3 (Fig. 2c–f). Treatment with both the bacteria and phages specific to them resulted in lower levels of cGAMP, STING and phospho-IRF3 relative to *S. enterica*-treated chickens, however, further elevation of non-modified IRF3 levels (relative to the birds infected with the bacteria) (Fig. 2c–f). These results corroborated the conclusion presented in the previous paragraph.

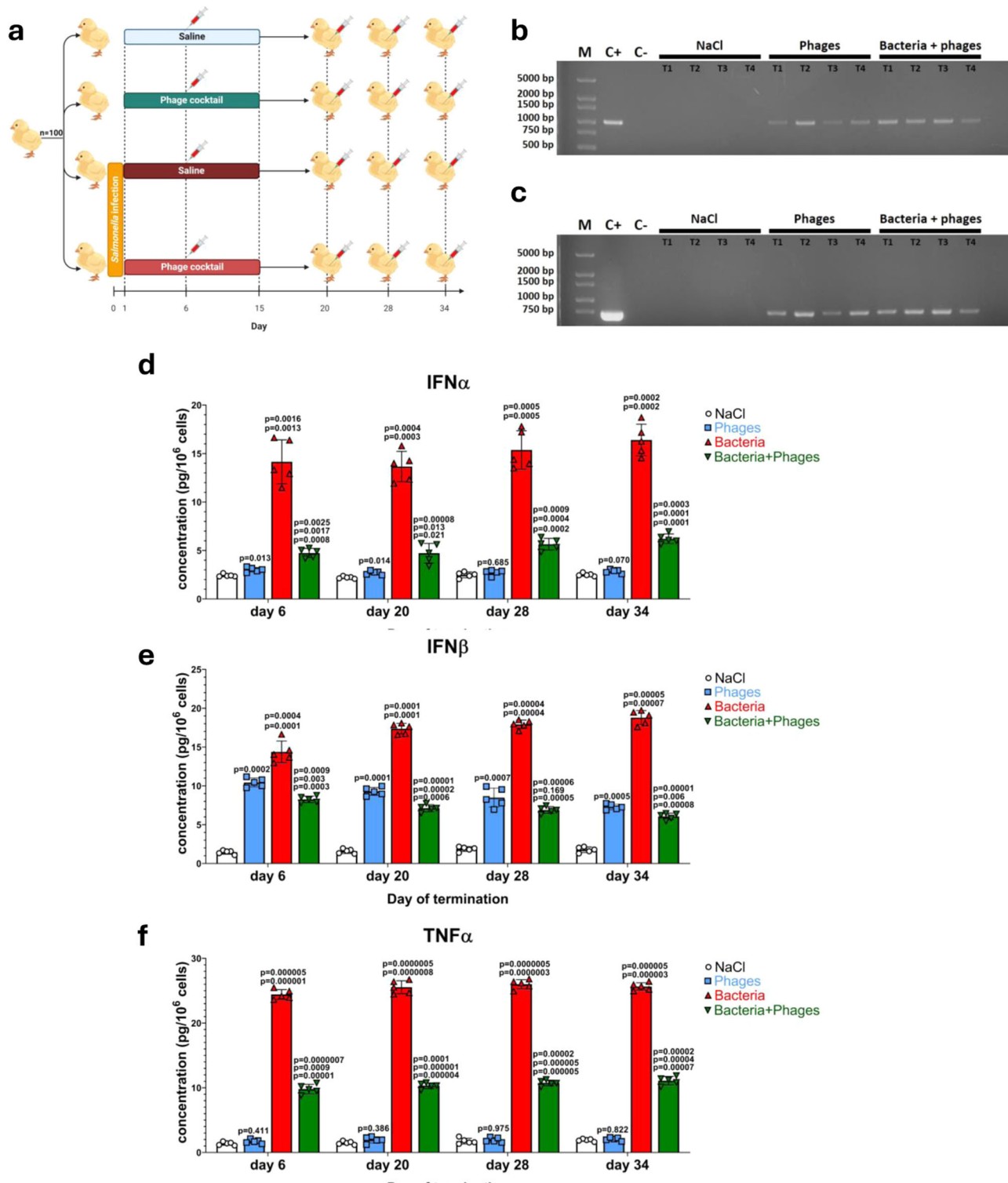

## The mechanism of blocking the IRF3 phosphorylation in response to bacteriophage DNA presence

Since the inhibition of IRF3 phosphorylation appeared to be crucial in blocking the immune response of chicken cells to the presence of bacteriophage DNA, we aimed to investigate the molecular mechanism of this regulatory process. Earlier studies demonstrated that RNA polymerase III is able to transcribe cytosolic DNA, thus synthesizing 5′-triphosphate double-stranded RNA (dsRNA)[31,32]. Such dsRNA, after being recognized by RIG-I-like receptors (RLRs), creates a signal stimulating the adapter proteins (MAVS) which

together with RLRs contribute to the activation of TBK1 and IKKε which is necessary to IRF3 phosphorylation[33]. We hypothesized that bacteriophage DNA cannot be effectively transcribed by RNA polymerase III, as no sequences resembling eukaryotic promoters and specific AT-rich regions are present in their relatively short genomes (in contrast, eukaryotic viruses contain specific promoters, and bacterial genomes are large enough to contain random sequences resembling RNA polymerase III-specific promoters). If this hypothesis is true, in the response to bacteriophage DNA, RLRs-MAVS-mediated activation of TBK1 and IKKε might be ineffective,

**Fig. 1 | The scheme of the experimental procedure and levels of IFN and TNF factors and phage DNA throughout the experiment.** The experiment (**a**) involved 100 chickens, divided into four groups (*n* = 25 in each group). Groups 3 and 4 were infected orally with *S. enterica* serovar Typhimurium strain 13 at day 0 (samples taken at this time (all groups) were for quality control only) at the dose of $10^6$ CFU/ml while groups 1 and 2 were not infected but treated orally with 1 ml of 0.89% NaCl. Then, starting on day 1, groups 1 and 3 received 1 ml of 0.89% NaCl daily for 2 weeks, while groups 2 and 4 received 1 ml of the phage cocktail (a mixture of bacteriophages vB_SenM-2 and vB_Sen-TO17, with the titer of $10^9$ PFU/ml each) for the same period. Blood samples for analyses were withdrawn on days 6, 20, 28, and 34. DNAs of bacteriophages vB_SenM-2 (**b**) and vB_Sen-TO17 (**c**) were detected by PCR with primers specific to selected sequences of phage genomes (representative electropherograms are shown, with lane M containing molecular weight markers (with DNA sizes shown on the left), and lanes C+ and C− indicating controls with and without purified phage DNA, respectively, and each reaction was repeated 5 times giving the same results). The blood samples were also treated to liberate proteins and measure levels of selected polypeptides with ELISA. Abundances of IFNα (**d**), IFNβ (**e**), and TNFα (**f**) were determined in all samples. Results presented in (**d**–**f**) correspond to samples obtained from chickens (*n* = 5 for each subgroup) treated with NaCl (white columns with circles representing individual results), phages (blue columns with squares), bacteria (red columns with triangles), and bacteria and phages (green columns with inverted triangles), and are mean values with error bars representing SD. For statistical analyses, the normality of the distribution of variables was checked with the Kolmogorov–Smirnov test and the homogeneity of the variances with Levene's test. Once both assumptions were met, the analysis was carried out using ANOVA and *post hoc* Tukey's test. Otherwise (one or both assumption(s) was/were not met), the Kruskal–Wallis test and *post hoc* Dunn test were applied (see the Source Data file for details). The *p* values obtained in the statistical analyses are indicated above the columns; the green columns (chickens treated with bacteria and phages) are marked with three numbers, indicating *p* values vs. the groups representing chickens treated with NaCl (lower value), phages (middle values), and bacteria (upper values); consequently, red and blue columns are marked with two and one *p* value(s), respectively, indicating comparisons with the results shown in columns located to the left of them on the panel. Statistically significant differences were considered when *p* < 0.05. The scheme (**a**) was created using BioRender.com (publication license no. AH26H-Q0OQ2). Detailed results demonstrated in this figure are included in the Source Data file.

resulting in impaired IRF3 phosphorylation and its inability to translocate to the nucleus.

To test the above hypothesis, we have used a line of chicken lymphoblasts (the DT40 line), transfected with either bacteriophage DNA (isolated from both vB_SenM-2 and vB_Sen-TO17) or genomic DNA from the *S. enterica* strain (non-transfected cells were employed as a negative control). We demonstrated that the efficiency of transfection by both phage genomes and the fragmented bacterial genome was high enough to indicate the efficient entrance of foreign DNAs into chicken cells (Fig. 4a). Moreover, the absence of considerable amounts of the endotoxin in all tested DNA samples was confirmed in the LAL test. The determined values were 0.87, 0.98, and 0.96 EU/ml for DNAs of vB_SenM-2, vB_Sen-TO17, and *S. enterica*, respectively, where values < 1 EU/ml are considered as ultra-pure samples[34].

To confirm that the response of cultured chicken lymphoblasts to phage and bacterial DNAs was comparable to that observed in the blood samples of birds infected with *S. enterica* and/or treated with the phage cocktail, we have determined the efficiency of expression of genes coding for cGAS, STING, and IRF3 using RT-qPCR. The obtained results confirmed that the tendencies of changes in the expression of tested genes determined in the transfected lymphoblasts were comparable to those observed in samples of chicken blood when levels of corresponding gene products were estimated (Fig. 4b). Moreover, we have measured levels of the phosphorylated form of IRF3 in the transfected lymphoblasts, to show that they are significantly elevated in the presence of bacterial DNA, but not in the presence of phage DNA (Fig. 4c). These results indicated that the immune response to genomic DNA of *S. enterica* and to phage DNA observed in the transfected lymphoblasts is comparable to that observed in chickens.

To test whether dsRNA can be synthesized by RNA polymerase III in the presence of either phage or bacterial DNA in chicken cells, we have used an antibody specific to dsRNA. The detection of specific signals was performed using fluorescent microscopy and observation of transfected cells, and the dot-blot procedure with the material obtained after cell lysis. When cells transfected with fragments of the *S. enterica* chromosome were investigated, intensive signals derived from dsRNA were observed in both types of experiments (fluorescent microscopy and dot-blot), contrary to non-transfected cells when the signals were at the background level (Fig. 5a–d). When an inhibitor of the RNA polymerase III was used (at the final concentration corresponding to the $IC_{50}$ value; this concentration was chosen to avoid a significant inhibition of the cell growth, otherwise observed in cultures incubated for the time of the whole experiment at higher concentrations), the intensities of dsRNA-specific signals were significantly (about 50%) lower, indicating that the detected dsRNA molecules were produced at least predominantly by this type of RNA polymerase

(Fig. 5a–d). Different results were obtained in experiments with chicken lymphoblasts transfected with phage DNA. In this case, no significant increase in the levels of dsRNA could be observed in both types of experiments, and the abundance of dsRNA in cells bearing phage DNA was statistically indistinguishable from that in control (non-transfected) cells (Fig. 5a–d). These results corroborated the proposed hypothesis that the block in the signal transduction in response to the presence of phage DNA, observed at the stage of phosphorylation of IRF3, is due to the inability of production of dsRNA by RNA polymerase III under these conditions, likely due to a lack of recognizable sequences in bacteriophage genomes.

To test whether the mechanism of blocking the cGAS-STING pathway is specific to chickens or it is a more common phenomenon, we have repeated the kinds of experiments described in the above paragraph with mouse splenocytes transfected with either phage or bacterial DNA. Splenocytes were isolated from wild-type mice which were tested for their general health conditions to confirm the consistency of all parameters with the norms (Supplementary Table S1).

We confirmed that similar to chickens, mouse splenocytes responded to the presence of bacterial DNA by enhanced production of both IFNα and IFNβ, while only IFNβ (but not IFNα) was produced after transfection of these cells with bacteriophage DNA (Supplementary Fig. S4). Since in the case of mouse splenocytes the transfection efficiencies by bacterial and phage DNAs were similar to those observed for chicken lymphoblasts, and modulations of levels of selected proteins involved in the immune response were comparable between mouse and chicken cells (Supplementary Fig. S5), we concluded that responses of mammalian and bird cells to bacterial and phage DNAs are comparable, including the inhibition of the cGAS-STING signal transduction pathway at the stage of IRF3 phosphorylation in the presence of bacteriophage DNA. Therefore, we have tested the efficiency of dsRNA production by RNA polymerase III in response to the presence of bacterial and phage DNAs in mouse splenocytes. In these experiments, results similar to those obtained with chicken lymphoblasts were evident. Namely, dsRNA was efficiently produced in cells transfected with bacterial DNA, but not in those transfected with phage DNA (Supplementary Fig. S6). Therefore, we conclude that the immune responses to phage DNA, including the mechanism of the inhibition of the cGAS-STING pathway, are similar in both chickens and mice.

### The TLR9- and MyD88-dependent signal transduction pathway is responsible for the production of INFβ by blood cells of chickens treated with bacteriophages

Since the cGAS-STING pathway of the response of chicken's cells to the presence of bacteriophage DNA was found to be blocked at the stage of the IRF3 phosphorylation, we have tested the efficiency of the

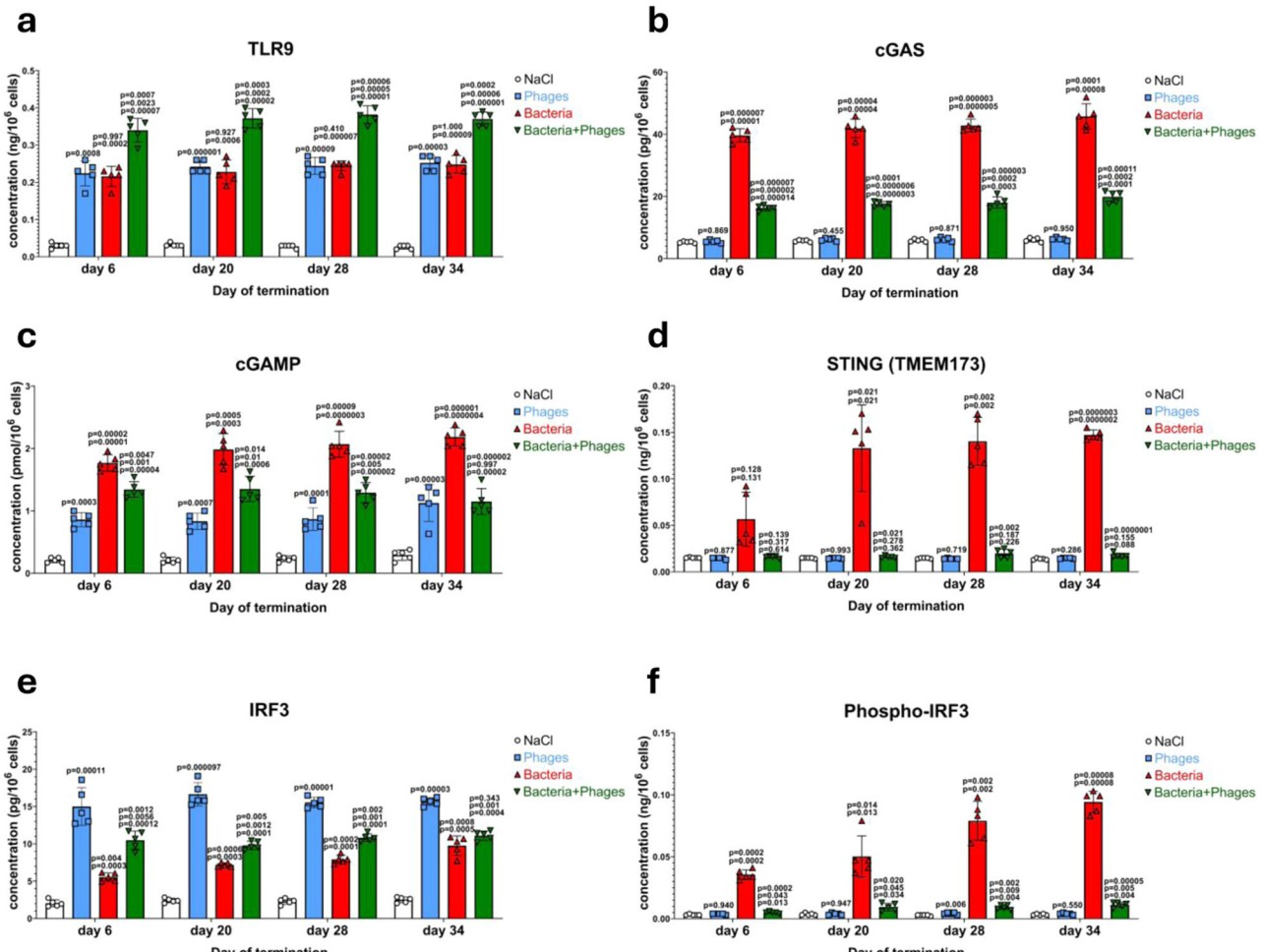

**Fig. 2 | Unlike activation of TLR9, the cGAS-STING signaling pathway, which is induced by bacteriophages, but not that induced by *S. enterica* infection in chickens, is blocked at the stage of IRF3 phosphorylation.** Levels of TLR9 (**a**), cGAS (**b**), cGAMP (**c**), STING (TMEM173) (**d**), IRF3 (**e**), and phosphorylated IRF3 (**f**) were determined by ELISA in samples of blood obtained as indicated in Fig. 1. Results correspond to samples obtained from chickens ($n = 5$ for each subgroup) treated with NaCl (white columns with circles representing individual results), phages (blue columns with squares), bacteria (red columns with triangles), and bacteria and phages (green columns with inverted triangles), and are mean values with error bars representing SD. For statistical analyses, the normality of the distribution of variables was checked with the Kolmogorov–Smirnov test and the homogeneity of the variances with Levene's test. Once both assumptions were met,

the analysis was carried out using ANOVA and *post hoc* Tukey's test. Otherwise (one or both assumption(s) was/were not met), the Kruskal–Wallis test and *post hoc* Dunn test were applied (see the Source Data file for details). The $p$ values obtained in the statistical analyses are indicated above the columns; the green columns (chickens treated with bacteria and phages) are marked with three numbers, indicating $p$ values vs. the groups representing chickens treated with NaCl (lower value), phages (middle values), and bacteria (upper values); consequently, red and blue columns are marked with two and one $p$ value(s), respectively, indicating comparisons with the results shown in columns located to the left of them on the panel. Statistically significant differences were considered when $p < 0.05$. Detailed results demonstrated in this figure are included in the Source Data file.

second pathway, dependent on the activation of the TLR9 receptor (shown to be stimulated by phage DNA as indicated in Fig. 2a). A strong induction of abundance of the subsequent factors in this pathway, MyD88 (Fig. 6a), IRAK1 (Fig. 6b), IRAK4 (Fig. 6c), and TRAF6 (Fig. 6d), was evident in the blood cells of the phage cocktail-treated chickens. The levels of proteins operating in the next steps of this signal transduction process, TBK1 (and its phosphorylated form) and IKKε, were also elevated under these conditions, similarly to the factors included in the complex activating these kinases, TANK, TBKBP1, and NAP1 (Fig. 3a–f; these factors are common in both TLR9/MyD88 and cGAS-STING pathways).

The last step of the signal transduction pathway dependent on TLR9 and MyD88 is the activation of two transcription factors, IRF7 and NF-κB. Importantly, IRF7 is activated by phosphorylation, while NF-κB is stimulated by dephosphorylation, thus, only phosphorylated IRF7 and non-phosphorylated NF-κB can be translocated from the cytoplasm to the nucleus and stimulate transcription of specific genes,

between others that coding for IFNβ. The results demonstrated in Fig. 7a, b indicate that levels of both IRF7 and its phosphorylated form were increased in samples of blood of chickens treated with the phage cocktail, demonstrating that activation of this transcription factor occurred indeed. Moreover, despite the occurrence of low levels of the phosphorylated form of NF-κB under these conditions (Fig. 7c), its dephosphorylated form was especially highly abundant in the tested samples (Fig. 7d). These results provided evidence that IRF7 and NF-κB were efficiently activated upon administration of the phage cocktail to chickens which results in effective production of IFNβ. Similar results were obtained in experiments with chicken lymphoblasts (Fig. 4d–f), further corroborating the compatibility of both experimental systems.

The stimulation of the tested signal transduction pathway occurred also after infection of chickens by *S. enterica* (Figs. 6 and 7). However, contrary to the cGAS- and STING-dependent pathway, the addition of the phage cocktail further stimulated, rather than inhibited, the host response to the bacterial infection.

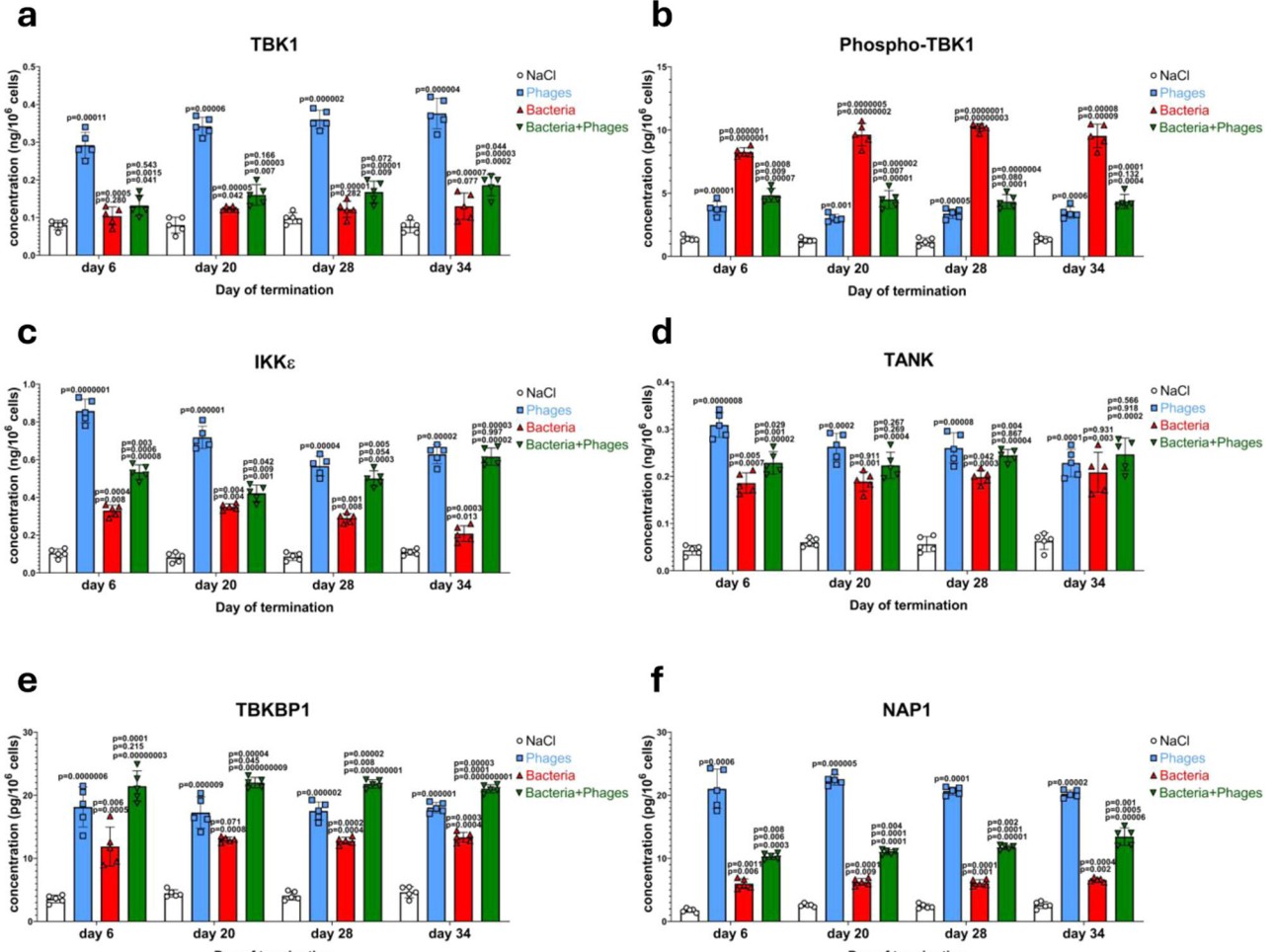

**Fig. 3 | Levels of proteins involved in both cGAS-STING and TLR9/MyD88 signaling pathways are elevated by both bacteriophages and *S. enterica* infection in chickens.** Levels of TBK1 (**a**), phosphorylated TBK1 (**b**), IKKε (**c**), TANK (**d**), TBKBP1 (**e**), and NAP1 (**f**) were determined by ELISA in samples of blood obtained as indicated in Fig. 1. Results correspond to samples obtained from chickens (*n* = 5 for each subgroup) treated with NaCl (white columns with circles representing individual results), phages (blue columns with squares), bacteria (red columns with triangles), and bacteria and phages (green columns with inverted triangles), and are mean values with error bars representing SD. For statistical analyses, the normality of the distribution of variables was checked with the Kolmogorov–Smirnov test and the homogeneity of the variances with Levene's test. Once both assumptions were met, the analysis was carried out using ANOVA and *post hoc* Tukey's test. Otherwise (one or both assumption(s) was/were not met), the Kruskal–Wallis test and *post hoc* Dunn test were applied (see the Source Data file for details). The *p* values obtained in the statistical analyses are indicated above the columns; the green columns (chickens treated with bacteria and phages) are marked with three numbers, indicating *p* values vs. the groups representing chickens treated with NaCl (lower value), phages (middle values), and bacteria (upper values); consequently, red and blue columns are marked with two and one *p* value(s), respectively, indicating comparisons with the results shown in columns located to the left of them on the panel. Statistically significant differences were considered when *p* < 0.05. Detailed results demonstrated in this figure are included in the Source Data file.

## Discussion

The question about possible interactions between bacteriophages and eukaryotic organisms appeared important as it has been recognized that these viruses are the most abundant biological entities on Earth, that they occur in large numbers in human/animal gut, and that they might be used to combat bacterial infections (the re-discovery of the primary findings made over a century ago)[2–12]. Although bacteriophages are obligatory parasites of bacteria[1], recent studies provided evidence that they influence not only the microbiome of human/animal gut but also affect the eukaryotic host immune system[13,14,18–20].

It is obvious that proteins of bacteriophage capsids, like any foreign proteins introduced into human/animal organisms, can induce the adaptive immune response[35]. However, the indication that they may also induce an antiviral immune response was rather surprising. The latter discovery is exemplified by the research on a filamentous bacteriophage Pf, specific to *Pseudomonas aeruginosa*[26]. This phage, bearing a single-stranded DNA genome and influencing the virulence and antibiotic resistance of its bacterial host[36,37], can be internalized by mammalian cells through endocytosis which leads to the appearance of phage-derived RNA molecules that are recognized by TLR3; this initiates the pathway resulting in production of type I interferon[26].

Apart from the recognizing phage RNA by TLR3, DNA originating from bacteriophage genomes can be recognized by TLR9[27,38], as this receptor detects DNA with unmethylated CpG sequences which are common in both bacteriophage genomes and genomes of viruses and bacteria infecting human/animal cells[39–42]. Still, molecular mechanisms of the response of human/animal cells to bacteriophage DNA through the TLR9 pathway are still relatively poorly understood. Contrarily, to our knowledge, recognition of bacteriophage DNA by the second major system of the antiviral response, resulting in the stimulation of interferon production and based on sensing cytosolic DNA, the cGAS-STING-dependent pathway[43–45], has never been reported in the literature. In this light, we aimed to investigate mechanisms of bacteriophage recognition by animal cells, especially through the above mentioned two major pathways, dependent on the TLR9/MyD88 and

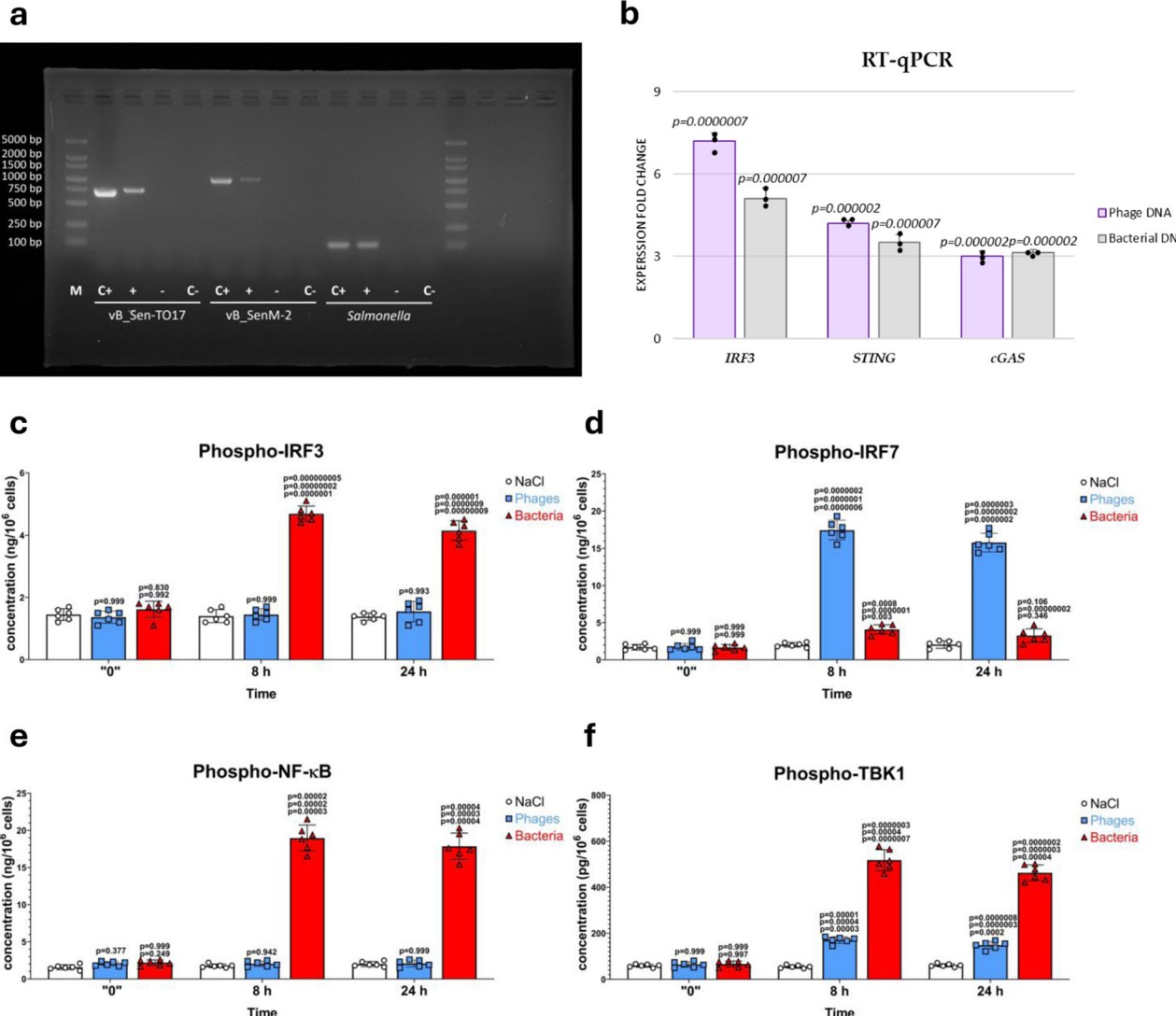

**Fig. 4 | The immune response of chicken lymphoblasts to phage and bacterial DNAs is similar to that of chickens infected with either phages or *S. enterica*.** The efficiency of transfection of chicken lymphoblasts by DNA from bacteriophages vB_Sen-TO17 and vB_SenM-2, and from *S. enterica* serovar Typhimurium strain 13 was confirmed by PCR using DNA isolated from transfected cells, followed by agarose gel electrophoresis (**a**) (representative electropherograms are shown, with lane M containing molecular weight markers (with DNA sizes shown on the left), and lanes C+ and C− indicating controls with and without purified phage/bacterial DNA, respectively, and (+) and (−) indicating transfected and untransfected cells, respectively; each reaction was repeated 3 times giving the same results). Levels of mRNA of genes coding for IRF3, STING, and cGAS in chicken lymphoblasts 24 h after transfection with phage or bacterial DNA were determined by RT-qPCR, and are shown as fold-change relative to the untransfected cells (**b**). Levels of phosphorylated IRF3 (**c**), phosphorylated IRF7 (**d**), phosphorylated NF-κB (**e**), and phosphorylated TBK1 (**f**) were determined by ELISA in lysates of chicken lymphoblasts 24 h after transfection with phage or bacterial DNA. In (**b**), results correspond to mRNA levels in cells transfected with either phage or bacterial DNA, relative to those without transfection (*n* = 3; independent biological repeats), and represent mean values with dots showing individual results, and error bars indicating SD. In (**c**–**f**), results correspond to samples obtained from cells (*n* = 6; independent biological repeats) treated with NaCl (white columns with circles representing individual results), phage DNA (blue columns with squares), and bacterial DNA (red columns with triangles), and are mean values with error bars representing SD. For statistical analyses (**b**–**f**), the normality of the distribution of variables was checked with the Kolmogorov–Smirnov test and the homogeneity of the variances with Levene's test. Once both assumptions were met, the analysis was carried out using ANOVA and *post hoc* Tukey's test. Otherwise (one or both assumption(s) was/were not met), the Kruskal–Wallis test and *post hoc* Dunn test were applied (see the Source Data file for details). The *p* values obtained in the statistical analyses are indicated above the columns. In (**b**), they indicate results of comparison between transfected and non-transfected cells. In (**c**–**f**), columns are marked with numbers, indicating *p* values vs. the cells treated with NaCl (lower value), phage or bacterial DNA (middle values), and time 0 (upper values); consequently, particular columns are marked with one, two or three numbers. Statistically significant differences were considered when *p* < 0.05. Detailed results demonstrated in this figure are included in the Source Data file.

cGAS-STING systems, in the previously established[29,30], experimental model of chickens infected/uninfected with *S. enterica* and treated/untreated with the cocktail of two bacteriophages (bearing double-stranded DNA genomes) specific to this bacterium. Such studies were substantiated by the fact that enhanced production of IFNβ (but not of IFNα, IFNγ or TNFα) was detected in response to the administration of the phage cocktail to chickens (Fig. 1).

Although the levels of cGAS were not elevated in response to the presence of phage DNA, it was evident that the signal nucleotide cGAMP (cyclic GMP-AMP) was efficiently produced under these conditions (Fig. 2). However, it is the activity of cGAS (stimulated by DNA binding) rather than its level which is crucial for cGAMP synthesis. Therefore, this is the first demonstration of the activation of the cGAS-STING system after administration of bacteriophages to animals. Such

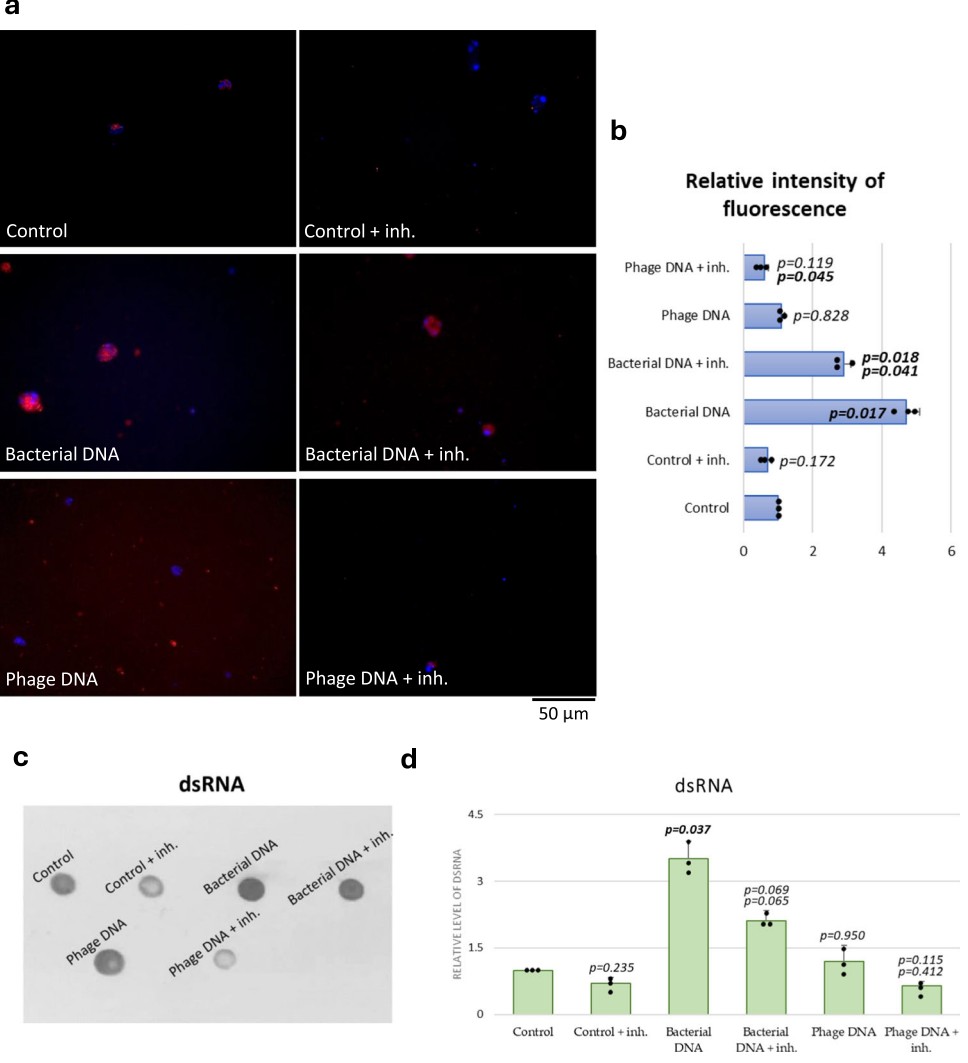

**Fig. 5 | The RNA polymerase III-synthesized dsRNA is abundant in chicken lymphoblast after transfection with bacterial DNA, but not after transfection with phage DNA.** The abundance of dsRNA in chicken lymphoblasts 24 h after transfection with bacterial (*S. enterica* serovar Typhimurium strain 13) or phage DNA was estimated by fluorescent microscopy (**a**, **b**) or dot-blot (**c**, **d**), using specific anti-dsRNA antibody (**a**–**d**) and DAPI staining (**a**). The inhibitor of RNA polymerase III was added to a final concentration of 25 μM (close to the $IC_{50}$ value for the human enzyme, estimated as 27 μM) immediately before transfection. Examples of micrographs (**a**) and blots (**c**) are presented, and quantification of the results based on analyses of 100 randomly selected cells, repeated 3 times as independent biological experiments (**b**) or 3 independent dot-blot experiments (**d**) is presented as mean values (from 3 values of independent experiments, shown as dots) with error bars representing SD. For statistical analyses (**b**, **d**), the normality of the distribution of variables was checked with the Kolmogorov–Smirnov test and the homogeneity of the variances with Levene's test. Once both assumptions were met, the analysis was carried out using ANOVA and *post hoc* Tukey's test. Otherwise (one or both assumption(s) was/were not met), the Kruskal–Wallis test and *post hoc* Dunn test were applied (see the Source Data file for details). The *p* values obtained in the statistical analyses are indicated above the columns. When one number is shown, it corresponds to the statistical analysis vs. control. When two numbers are presented (in experiments with DNA-transfected cells treated with the RNA polymerase III inhibitor), the upper one corresponds to the statistical analysis vs. control, and the lower one to statistical analysis vs. analogous experiments without the inhibitor. Statistically significant differences were considered when *p* < 0.05. Detailed results demonstrated in this figure are included in the Source Data file.

a conclusion is supported by the finding highly increased levels of IRF3 (Fig. 2), the transcription factor which stimulates the expression of genes encoding IFNβ and other cytokines. However, phosphorylation of IRF3 was impaired as levels of the phospho-IRF3 form were very low in blood cells of chickens treated with bacteriophages (Fig. 2). Hence, we suggested that DNA of bacteriophages which were internalized into chicken cells (thus occurring in the cytoplasm) could be sensed by the cGAS-STING system, however, the final step of the specific signal transduction pathway (phosphorylation of IRF3) was blocked, impairing the complete anti-viral response.

In this light, we aimed to elucidate what is the mechanisms of the impairment of the IRF3 activation when the cGAS-STING pathway is induced by bacteriophage DNA, contrary to the fully effective induction by viruses replicating in animal cells or by bacterial pathogens, like *S. enterica*, as demonstrated in this report (Fig. 2). Such a mechanism might be especially intriguing, as levels of TBK1 and its phosphorylated form, as well as levels of IKKε, the kinases necessary for IRF3 phosphorylation[33,46], were increased in blood cells of phage cocktail-treated chickens (Fig. 3). The same effects were observed for proteins included in the complex activating these kinases, TANK, TBKBP1, and NAP1 (Fig. 3).

One possible explanation might be that contrary to pathogens infecting eukaryotic cells, bacteriophages are unable to multiplicate in such hosts, as regulatory sequences present in their genomes cannot be recognized by specific eukaryotic enzymes. As demonstrated previously, in the cGAS-STING pathway, the RNA polymerase III

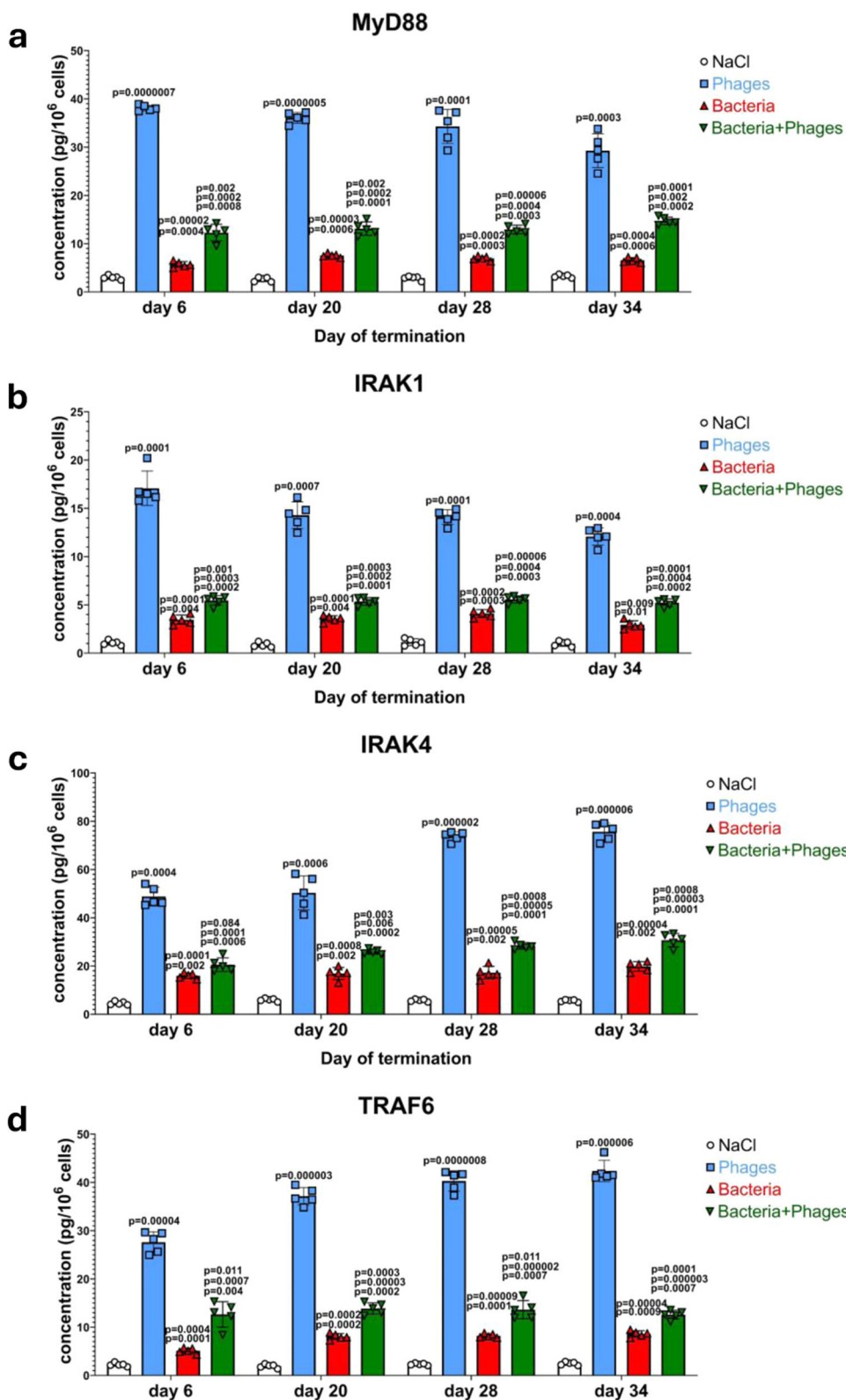

transcribes cytosolic DNA to produce dsRNA which is recognized by RIG-I-like receptors (RLRs)[31,32]. This is a signal to stimulate their adapter proteins, MAVS. Importantly, RLRs and MAVS are required to activate TBK1 and IKKε[33]. Therefore, we hypothesized that bacteriophage DNA cannot be effectively transcribed by RNA polymerase III, thus the RLRs-MAVS activation of TBK1 and IKKε is ineffective, causing impairment of the IRF3 phosphorylation and its subsequent translocation into the nucleus. Although early reports[47,48], as well as more recent studies[26], demonstrated the appearance of phage DNA-derived transcripts in eukaryotic cells, it is likely that such RNAs are products of random transcription from incidentally recognized promoter-like sequences, perhaps by one of eukaryotic RNA polymerases. Transcription

**Fig. 6 | The TLR9/MyD88 signaling pathway is induced by both bacteriophages and *S. enterica* infection in chickens.** Levels of MyD88 (**a**), IRAK1 (**b**), IRAK4 (**c**), and TRAF6 (**d**) were determined by ELISA in samples of blood obtained as indicated in Fig. 1. Results correspond to samples obtained from chickens ($n = 5$ for each subgroup) treated with NaCl (white columns with circles representing individual results), phages (blue columns with squares), bacteria (red columns with triangles), and bacteria and phages (green columns with inverted triangles), and are mean values with error bars representing SD. For statistical analyses, the normality of the distribution of variables was checked with the Kolmogorov–Smirnov test and the homogeneity of the variances with Levene's test. Once both assumptions were met, the analysis was carried out using ANOVA and *post hoc* Tukey's test. Otherwise (one or both assumption(s) was/were not met), the Kruskal–Wallis test and *post hoc* Dunn test were applied (see the Source Data file for details). The *p* values obtained in the statistical analyses are indicated above the columns; the green columns (chickens treated with bacteria and phages) are marked with three numbers, indicating *p* values vs. the groups representing chickens treated with NaCl (lower value), phages (middle values), and bacteria (upper values); consequently, red and blue columns are marked with two and one *p* value(s), respectively, indicating comparisons with the results shown in columns located to the left of them on the panel. Statistically significant differences were considered when $p < 0.05$. Detailed results demonstrated in this figure are included in the Source Data file.

catalyzed by RNA polymerase III and leading to the formation of RNA species activating the RLRs-MAVS system requires the presence of specific AT-rich sequences in the DNA template[31]. Moreover, it also requires the TATA-binding protein (TBP), a transcription factor cooperating with RNA polymerase III and responsible for its specificity. Indeed, among natural DNA sources, double-stranded DNA genomes of adenovirus and herpes simplex virus 1 (HSV-1), as well as fragments of the genome of *Legionella pneumophila* (a bacterial pathogen of humans), could induce the RLRs-dependent response, while various synthetic DNAs (lacking AT-rich regions), as well as calf thymus DNA, failed to activate IRF3 and to induce IFNβ production[31].

We suspected that bacteriophage DNA might be hardly recognized by RNA polymerase III, especially due to its compact nature, lacking stretches of AT-rich sequences, and the fact that bacteriophage promoters evolved to be recognized by bacterial RNA polymerase, not eukaryotic transcription factors. We tested this hypothesis by transfecting chicken lymphoblasts with either phage DNA or fragments of the *S. enterica* chromosome and monitoring the appearance of dsRNA molecules. Strong signals derived from dsRNA-specific antibody could be observed only in lymphoblasts transfected with bacterial DNA, but not those transfected with phage DNA (in the latter case, the signals were at the level of the background observed in non-transfected cells) (Figs. 4 and 5). Appearance of dsRNA-specific signals was impaired in the presence of the RNA polymerase III inhibitor, further corroborating the tested hypothesis (Fig. 5). Hence, we conclude that effective production of 5′-triphosphate dsRNA by eukaryotic RNA polymerase III, recognized by RLRs, is inefficient on the bacteriophage DNA template, eventually resulting in the block of the cGAS-STING pathway at the final step (phosphorylation of IRF3). The same results were obtained when mouse splenocytes were transfected with either bacterial or phage DNAs and the presence of dsRNA molecules was tested, indicating that the proposed explanation is not specific to chicken cells, but it is likely common to mammalian and bird immune systems (Supplementary Figs. S5 and S6). Therefore, the proposed mechanism can explain the block of the specific anti-viral response of eukaryotic cells to the presence of bacteriophage DNA.

Contrary to the cGAS-STING pathway of the cytosolic DNA sensors, the TLR9/MyD88-dependent pathway, stimulated by foreign DNA engulfed during the endocytosis, appears to be fully activated by bacteriophage DNA. We have observed stimulation of all crucial factors operating in this system (Figs. 2 and 3), and activation of two transcription factors involved in the expression of the gene coding for IFNβ, namely phosphorylation of IRF7 and dephosphorylation of NF-κB (Figs. 4 and 7). This leads to effective production of IFNβ (Fig. 1e).

Bacteriophage-induced production of IFNβ was found to be accompanied by elevated levels of anti-inflammatory cytokines (IL10 and IL4), but not pro-inflammatory interleukins[30]. In fact, IFNβ has been demonstrated to stimulate the production of IL10[49], and discussed for the importance of this activity[50]. This may explain a lack of a strong inflammatory response in animals and humans to the administration of bacteriophages.

The results of research on the effects of administration the phage cocktail to chickens were corroborated by those of

experiments with *S. enterica* infection, as well as with such an infection and phage therapy. As expected, the bacterial infection of chickens induced effectively both investigated pathways, cGAS-STING- and TLR9/MyD88-dependent. It appears that *S. enterica* DNA might be as effective in the stimulation of the RLRs-MAVS system as that of *L. pneumophila*, thus allowing eventual phosphorylation of IRF3 and induction of the specific response. Since IRF3 regulates the expression of thousands of genes[51], including those involved in the immune response, a positive feedback phenomenon may result in a further increase in the levels of many factors involved in the system described above, which is not the case when only bacteriophages are administered. Moreover, contrary to treatment with phages, the infection with *S. enterica* induced not only TLR9 but also other TLRs, like TLR3, TLR4, and TLR6 (Supplementary Fig. S3). Treatment of the *S. enterica*-infected chickens with the phage cocktail resulted in the impairment of the cGAS-STING pathway and stimulation of the TLR9 pathway. It is, however, unclear whether this modulation of the animal response to bacterial infection by bacteriophages was due to the eradication of *S. enterica* by specific phages, as demonstrated previously[29], or by the independent effects of these prokaryotic viruses on the animal immune system.

Demonstration of the molecular mechanism by which bacteriophage DNA does not induce a strong antimicrobial immune response through the cGAS-STING pathway, together with the documented production of anti-inflammatory cytokines in the presence of bacteriophage DNA, can explain the safety of administration of phages to animals, described previously[30]. This can be important in light of introduction of the phage therapy into veterinary medicine. It is worth mentioning that bacteriophages may be considered for their use as either feed supplements (additives) or therapeutics. Recent regulations of the European Union and European Medicines Agency defined the requirements for both these types of potential applications. The approval of bacteriophages as feed supplements (additives) is possible under Regulation (EC) No. 1831/2003 of the European Parliament and of the Council of September 22, 2003, on additives for use in animal nutrition. Nevertheless, on October 23, 2023, the European Medicines Agency's Regulation EMA/CVMP/NTWP/32862/2022 came into force on guidelines for the quality, safety and efficacy of veterinary medicinal products specifically for phage therapy. The requirements are significantly stricter and more difficult to fulfill than those for feed supplements, including safety issues. Therefore, studies like the work described in this report are important in the way that may lead to approval of the use of bacteriophages in the prevention or treatment of animals against bacterial infections.

In summary, this report demonstrates for the first time that bacteriophages bearing dsDNA genomes and administered orally to animals at a high dose ($10^9$ virions) induce the cGAS-STING pathway which is, however, blocked at the last stage, the phosphorylation of the IRF3 factor. The proposed mechanism of this blockage is based on the inability of RNA polymerase III to recognize bacteriophage DNA (likely due to the lack of specific promoter and/or specific AT-rich sequences, in contrast to DNAs of eukaryotic viruses or bacterial chromosomes; the latter being large enough to contain (by chance) sequences

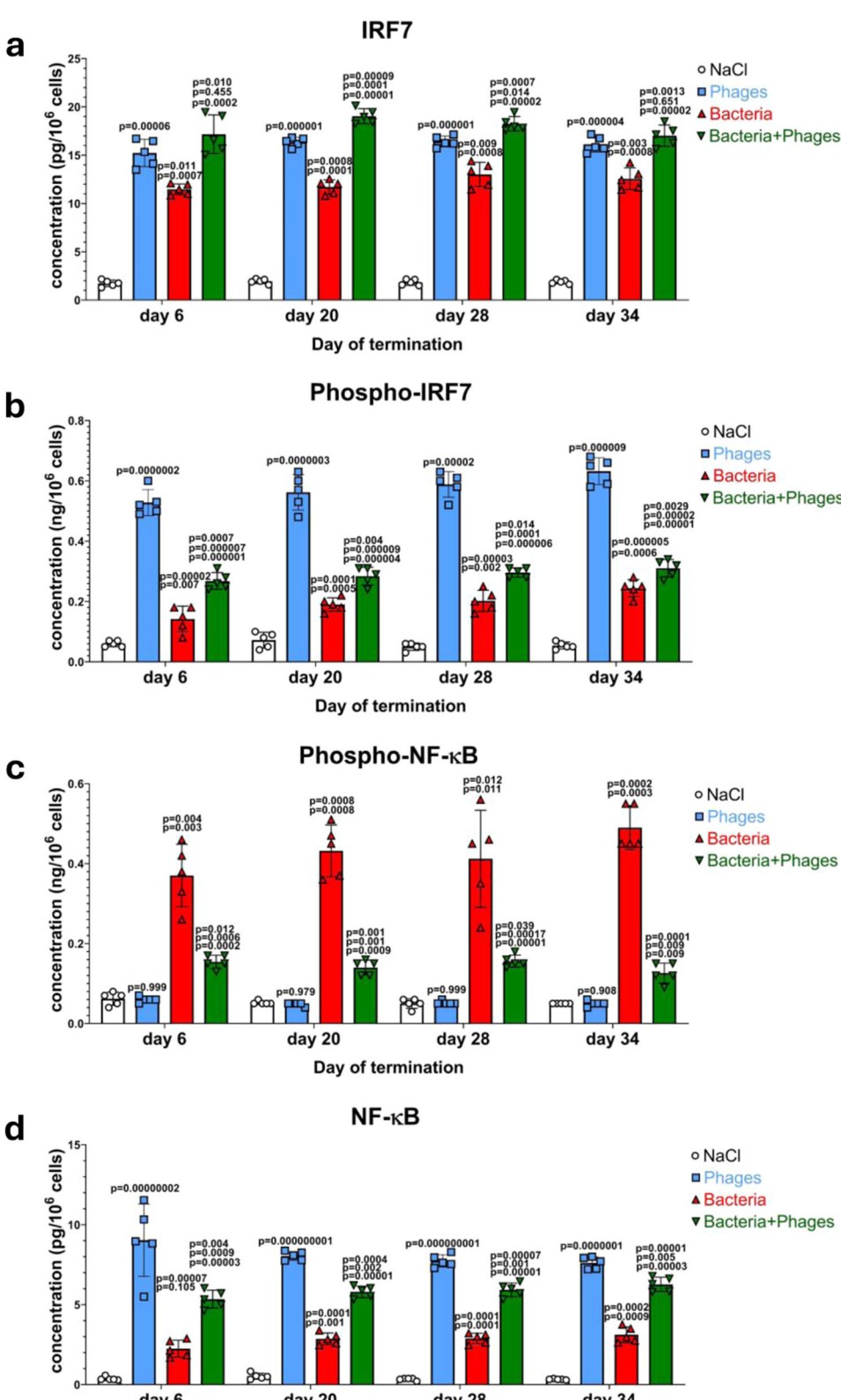

resembling RNA polymerase III-specific promoters and containing AT-rich stretches) and the resultant absence of dsRNA which is otherwise necessary to stimulate IRF3 phosphorylation. However, the TLR9-dependent pathway is fully activated by the presence of bacteriophage DNA, leading to activation of the IRF7 and NF-κB transcription factors and resultant production of IFNβ and anti-inflammatory cytokines. This provides the principle of the molecular mechanisms of the immune response of animals to bacteriophages, especially in detecting phage dsDNA and subsequent signaling pathways. The proposed model is shown schematically in Fig. 8.

**Fig. 7 | IRF7 and NF-κB transcription factors are activated after both administration of bacteriophages and *S. enterica* infection in chickens.** Levels of IRF7 (**a**), phosphorylated IRF7 (**b**), phosphorylated NF-κB (**c**), and NF-κB (**d**) were determined by ELISA in samples of blood obtained as indicated in Fig. 1. Results correspond to samples obtained from chickens ($n = 5$ for each subgroup) treated with NaCl (white columns with circles representing individual results), phages (blue columns with squares), bacteria (red columns with triangles), and bacteria and phages (green columns with inverted triangles), and are mean values with error bars representing SD. For statistical analyses, the normality of the distribution of variables was checked with the Kolmogorov–Smirnov test and the homogeneity of the variances with Levene's test. Once both assumptions were met, the analysis was

carried out using ANOVA and *post hoc* Tukey's test. Otherwise (one or both assumption(s) was/were not met), the Kruskal–Wallis test and *post hoc* Dunn test were applied (see the Source Data file for details). The $p$ values obtained in the statistical analyses are indicated above the columns; the green columns (chickens treated with bacteria and phages) are marked with three numbers, indicating $p$ values vs. the groups representing chickens treated with NaCl (lower value), phages (middle values), and bacteria (upper value); consequently, red and blue columns are marked with two and one $p$ value(s), respectively, indicating comparisons with the results shown in columns located to the left of them on the panel. Statistically significant differences were considered when $p < 0.05$. Detailed results demonstrated in this figure are included in the Source Data file.

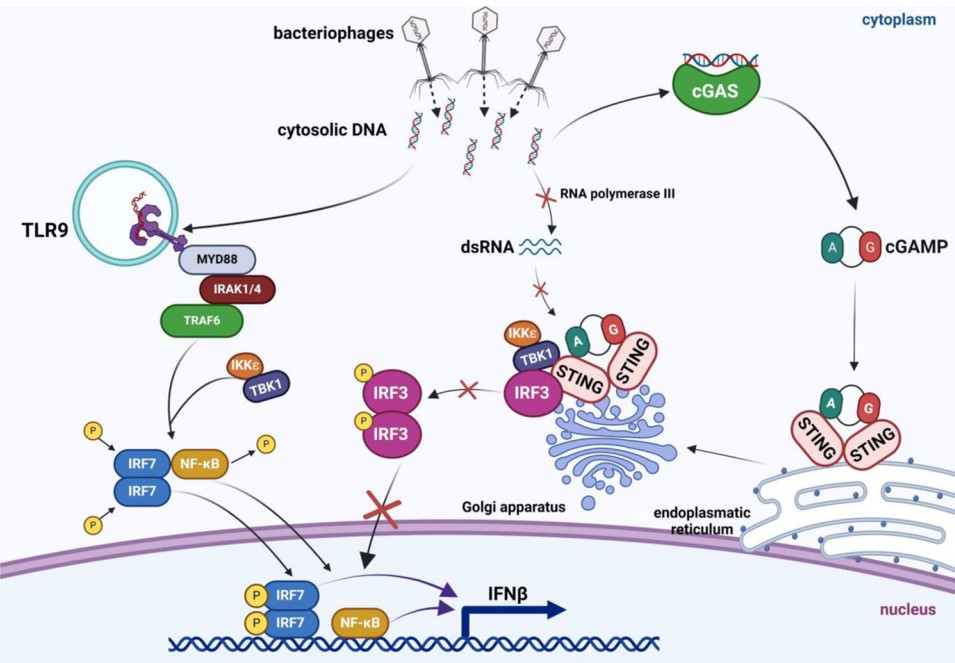

**Fig. 8 | The proposed model of the recognition of bacteriophage DNA by the animal immune system.** Bacteriophage DNA is proposed to be recognized by two systems, cGAS-STING and TLR9/MyD88. While the latter system works efficiently, leading to activation of IRF7 and NF-κB transcription factor, and subsequent production of INFβ, the cGAS-STING-dependent pathway is blocked at the stage of IRF-3 phosphorylation. This impairment of IRF3 phosphorylation results from a lack of

stimulation of this process by TBK1 and IKKε (in cooperation with TANK, TBKBP1, and NAP1) due to ineffective RLRs-MAVS activation in response to a lack of RNA polymerase III-catalyzed transcription of bacteriophage DNA and impaired production of dsRNA, because of the absence of specific AT-rich sequences and/or eukaryotic-like promoters in phage genomes. The scheme was created using BioRender.com (publication license no. BQ26HQ0INZ).

## Methods

### Animals

The experiments with chickens (*Gallus gallus domesticus*) were conducted using birds (not genetically modified) which were delivered by the breeder (national registration number: PL28036602). All procedures were approved by the Local Ethical Committee for Experimental Animals in Olsztyn, Poland (permission 62/2019 dated July 30, 2019).

During the experiments, the animals were housed in the Experimental Bird Infection Pavilion at the Department of Bird Diseases, Faculty of Veterinary Medicine, University of Warmia and Mazury in Olsztyn, Poland. The pavilion was adequately provided with a pressure cascade in the boxes, sluices and sanitary corridors, as well as a high-efficiency particulate absorption (HEPA) filter system to prevent contamination of the experimental rooms.

The birds were housed in 8 m² boxes with 25 animals in each. The following conditions were set in each box: 75% average humidity, forced ventilation of 17 air changes/h, and regular light/dark cycles (12 h day/12 h night, with a light intensity of 10 lx). The temperature was lowered from 33 °C at the beginning of the experiment to 22 °C at the end of the experiment. The animals were fed a complete feed *ad libitum* and had access to water 24 h a day.

The mouse (*Mus musculus*) line C57BL/6J (purchased from the Tri-City Central Animal Laboratory, Research and Service Center of the Medical University of Gdansk, Poland) was used. Animals were housed in a room with ventilation (15 air volume changes/h), in laboratory cages of 15 cm high and 400 cm² of total surface size. Artificial lighting allowed to keep the "day/night" conditions as 12 h light/12 h dark. The ambient temperature was maintained at $22 \pm 2\,°C$ and humidity at $50 \pm 5\%$. The access of mice to food and tap water was *ad libitum*. For euthanasia, mice were anesthetized with 2.5% isoflurane inhalation at the flow rate of 0.5 l/min; then 120 mg/kg pentobarbital was administered intraperitoneally. The experiments with mice were approved by the Local Ethics Committee for Experimental Animals in Bydgoszcz, Poland (permission number 02/2022, dated January 19, 2022).

### Bacteriophages and bacterial strains

In the experiments, two bacteriophages, vB_SenM-2 and vB_Sen-TO17, were used (both available from authors at request). They come from the collection of the Department of Molecular Biology, Faculty of Biology, University of Gdansk, and were described previously[52,53]. Both phages specifically infect strains of *S. enterica*. The bacteriophage vB_SenM-2 is a caudate phage, with a dsDNA genome of 158,986 bp

(GenBank accession no. KX171211). Its capsid diameter is 84 × 79 nm, and its tail length is 111 nm[52]. Bacteriophage vB_Sen-TO17 is a caudate phage having dsDNA as a genetic material with a size of 41,658 bp (GenBank accession no. MT012729). Its capsid diameter is 48 × 46 nm, and its tail length is 121 nm[53].

The strain of *S. enterica* serovar Heidelberg (a host for bacteriophage vB_SenM-2 propagation) came from the collection of the Department of Molecular Biology, University of Gdansk, Poland. The strain of *S. enterica* serovar Typhimurium 13 (a host for bacteriophage vB_Sen-TO17 propagation and the strain used for infection of chickens) was obtained from the National Salmonella Center at the Medical University of Gdansk, Poland. This strain is sensitive to both vB_SenM-2 and vB_Sen-TO17, as reported previously[29]. Both *S. enterica* strains were described elsewhere[35]. Isolation of *S. enterica* in chicken fecal samples and cloacal swabs was conducted in accordance with ISO 6579-1:2017 standards. Serotypes of the isolated bacteria were confirmed by serological identification using the SIT EnTy Kit from Immunolab (Gdansk, Poland).

## Preparation and purification of the phage cocktail

Phage lysates were prepared and purified according to the previously published protocol[30]. Bacteriophages were propagated in a chosen bacterial host (see subsection "Bacteriophages and bacterial strains") and then concentrated with polyethylene glycol 8000 (PEG 8000) (BioShop, Burlington, Ontario, Canada; at 10% final concentration of PEG 8000) overnight at 4 °C. The prepared lysates were centrifuged at $10,000 \times g$ for 30 min, at 4 °C (Avanti JXN-26, rotor JLA-8000, Beckman Coulter, Indianapolis, USA). The supernatants were removed, and the pellets were suspended in 0.89 % NaCl. PEG 8000 was then removed by adding 2 ml of chloroform and centrifugation at $4000 \times g$ for 15 min, at 4 °C (Avanti JXN-26, rotor JS-13.1, Beckman Coulter, Indianapolis, USA). The washing procedure was repeated until PEG8000 could not be observed. Lysates were ultracentrifuged in a CsCl (Merck, #C4036) concentration gradient at $95,000 \times g$ for 2.5 h, at 4 °C (Optima XPN-100, rotor SW32.1 Ti, Beckman Coulter, Indianapolis, USA) To remove residual sucrose, 2 ml of bacteriophage lysates were dialyzed against 300 ml of 0.15 M NaCl, using a dialysis membrane (ZelluTrans, MWCO: 12.000-14.000, serial number: E674.1; Roth, Germany) for 7 days at 4 °C. The NaCl solution was replaced every 12 h.

To exclude possible contamination of the lysates with bacterial endotoxins, their levels were checked using Pierce™ LAL Chromogenic Endotoxin Quantitation Kit (Thermo Fisher Scientific, #12117850). The levels were $0.156 \pm 0.062$ and $0.085 \pm 0.043$ EU/kg/h, for vB_SenM-2 and vB_Sen-TO17, respectively (with the negative control <0.05 EU/kg/h, and the upper acceptable limit of the endotoxin concentration equal to 5.0 EU/kg/h).

The phage lysates were mixed at a ratio of 1:1 v/v ($10^9$ PFU/ml each). Before administration to the animals, the cocktail was prepared in 20 mM $CaCO_3$ to neutralize the acidic pH in the stomach (this increased the survival of bacteriophages).

## Eukaryotic cells and cell cultures

The DT40 cell line of chicken lymphoblasts was purchased from the American Type Culture Collection (ATCC). Cells were cultured in Dulbecco's modified Eagle's medium (Sigma-Aldrich, # D6429). The medium contained 4 mM L-glutamine, 4.5 g/l glucose, 1.5 g/l sodium bicarbonate and 0.05 mM 2-mercaptoethanol (75% of the medium volume); tryptose phosphate broth (10% of the medium volume, Thermo Fischer, #18050039); fetal bovine serum (10% of the medium volume, Gibco, #10270106); and chicken serum (5% of the medium volume, Sigma-Aldrich, #C5405). Cells were incubated at 37 °C at 5% $CO_2$.

Mouse splenocytes were isolated from spleens of naïve, 9-month-old, C57BL/6J mice. The cells were cultured as described above or in the RPMI 1640 medium (Sigma-Aldrich, #R8758) containing 10% fetal bovine serum (Gibco, #10270106), 2 mM L-glutamine, Antibiotic-Antimycotic (Gibco, #15240062) in the presence of 5 µg/ml concanavalin A (ConA) (Sigma-Aldrich, #L6397,) in 12-well culture plates at 5% $CO_2$ and 37 °C for 24 h.

## Groups of chickens and the experimental scheme

Since the purpose of this work was to investigate the mechanism of the immune response to the genetic material of the bacteriophages, the experiments focused on four experimental groups. Each group consisted of 25 7-day-old chickens. The first two groups were controls that were not infected with bacteria. One of them was given physiological saline (0.89% NaCl) and the other received the phage cocktail from day 1 to day 15 of the experiment. The other two groups were infected with 1 ml of *S. enterica* serovar Typhimurium stain 13 ($10^6$ CFU/ml), suspended in 0.89% NaCl, into the beak, with the last group receiving 1 ml of the phage cocktail ($10^9$ PFU/ml of each phage) every day for 14 days, starting 24 h after the infection time. Blood samples were collected at four-time points by sacrificing 5 chickens from each group in a carbon dioxide chamber. Terminations were carried out on the 6th, 20th, 28th and 34th days of the experiment.

## Blood collection

Approximately 5 ml of blood was collected from every chicken. During the procedure, the birds were gently restrained to allow the needle (25-gauge, 1-in-long needle) to be inserted into the brachial wing vein at an angle between 10° and 20°. To avoid clot formation, both the needles and test tubes contained sodium heparin. The whole blood was subjected to morphological analysis, while the remainder was centrifuged to obtain plasma. In order to allow the determination of intracellular proteins, centrifugation parameters were chosen to ensure maximum disintegration of blood cells ($1800 \times g$ for 15 min at 4 °C). The plasma obtained by this method was stored at −80 °C until further analysis.

## Sandwich immunoenzymatic assay (ELISA)

All reagents and samples were brought to room temperature (21–25 °C) before analysis. Further assay steps followed the instructions provided by manufacturers of the commercial reagent kits (Shanghai Coon Koon Biotech Co., Ltd., Shanghai, China; Wuhan Xinqidi Biological Technology Co., Ltd, Wuhan, China; Enlibio, Wuhan, China; Cell Signaling Technology, Massachusetts, USA; Elabscience, Texas, USA; and MyBiosource, San Diego, USA). Standard curves for antibody validation are presented in Supplementary Fig. S7. For determination of levels of TLR4 (#EIA06452Ch), IFNβ (#CK-bio-27460), TNFα (#CK-bio-18240), TBKBP1 (#CK-bio-22537), cGAS (#CK-bio-27451), IKKε (#CK-bio-22816), IFI16 (#CK-bio-22673), IRF3 (#CK-bio-22057), phospho-IRF3 (Ser 396) (#CK-bio-29556), phospho-IRF7 (Ser 477) (#CK-bio-22459), and TANK (#CK-bio-25271), a 1:1 dilution was used (25 µl of the tested sample and 25 µl of the diluent). For determination of levels of IRAK1 (#CK-bio-27453), IRAK4 (#CK-bio-27452), MyD88 (#CK-bio-27461), DDX41 (#CK-bio-22537), IRF7 (#CK-bio-26639), NAP1 (#CK-bio-26749), cGMP (#CK-bio-24454), IFNα (#CK-bio-18168), NF-κB (#CK-bio-22695), phospho-NF-κB (S932) (#CK-bio-29204), phospho-TBK1 (Ser172) (#CK-bio-29739), TBK1 (#CK-bio-28646), TLR9 (#CK-bio-23737), TMEM 173/STING (#CK-bio-23792), TLR6 (#CK-bio-23254), TLR3 (#CK-bio-23730), and TRAF6 (#CK-bio-26795), samples were not diluted. For analyses of the DT40 cell line and mouse splenocytes, cells were washed three times with pre-cooled PBS. For each sample of $1 \times 10^6$ cells, 150–250 µl of the lysis buffer were added. The freeze-thaw process was repeated several times until the cells were fully lysed. Next, the cells were centrifuged for 10 min at $1500 \times g$ at 4°C and then the cell fragments were removed from the supernatant. In case of mouse splenocytes, phospho-IRF3 (Ser 379) (#54395), phospho-NF-κB (#E-EL-M0838) and TBK1 (#MBS2533500) levels were determined. Absorbance was measured at 10 min after stopping the reaction, using a Multiskan FC microplate reader

(Thermo Fisher Scientific, Massachusetts, USA), coupled with Skanlt 6.1.1. We used the RE software, which analyzes spectrophotometric color intensity, plots a standard curve based on the standards employed and reads the concentration values in the plasma samples tested. The results obtained were given in pg/ml or ng/ml (the minimum sensitivity of the test was 10 pg/ml or 0.1 ng/ml, respectively).

### Isolation of total DNA from blood samples, *Salmonella* cells, and phage lysates

The plasma of chickens treated with saline, treated with the phage cocktail, or infected with *S. enterica* and treated (or not) with the phage cocktail were used for isolation of total DNA. Plasma samples from five individuals in each group, at the respective time points, were pooled (50 μl from each individual). RNase A (final concentration 5 μg/μl; EURx, #E1350-01) was added to 250 μl of plasma and incubated for 30 min at 37 °C. The RNase was then inactivated for 10 min at 65 °C. Then, 5 μl of Proteinase K (final concentration 25 mg/ml; EURx, #E4350-01) and 400 μl Tissue Cell Lysis Solution (Lucigen, USA) were added and incubated for 30 min at 65 °C. The samples were cooled for 10 min on ice, and 300 μl of MPC Protein Precipitation Reagent (Lucigen, USA) were added. The solutions were centrifuged ($8000 \times g$, 10 min, 4 °C), and 500 μl of 100% isopropanol were added, followed by 24 h incubation at −20 °C. The samples were centrifuged ($9600 \times g$, 20 min, 4 °C), and supernatants were removed. The pellets were washed with 700 μl of 70% ethanol and centrifuged ($9600 \times g$, 40 min, 4 °C). The supernatants were removed, 500 μl of 70% ethanol was added again, and the samples were centrifuged ($9600 \times g$, 20 min, 4 °C). The supernatants were carefully removed, and the pellets were dried for 20 min at 30 °C under vacuum. Each pellet was resuspended in 30 μl of the nuclease-free water (Roth, Germany) and incubated at 37 °C until dissolution.

For isolation of genomic DNA from *Salmonella* cells and phage DNA from bacteriophages vB_SenM-2 and vB_Sen-TO17, the same procedure was employed, but cultures of *S. enterica* and CsCl density gradient-purified phage lysates were used instead of blood samples. The lack of significant amounts of endotoxins was confirmed using the Pierce™ LAL Chromogenic Endotoxin Quantitation Kit (Thermo Fisher Scientific, #12117850), as the measured values were <1 EU/ml.

### Primer design

The primers were designed for phage and bacterial DNA identification using PCR. Specific primers were designed by the Primer-BLAST software, with parameters set to exclude *Mus musculus* (taxid: 10090), *Gallus gallus* (taxid: 9031), Caudoviricetes (taxid: 2731619), and Viral (taxid: 10239) sequences. The primers (20 nucleotides length) are complementary to positions flanking the 37654–38634 bp region of bacteriophage vB_Sen-TO17 genome (included in the genes encoding hypothetical tail and neck proteins). For bacteriophage vB_SenM-2, primers were designed to target the nucleotide span 157621-157987 within the genome (included in the gene encoding a hypothetical neck protein). For *S. enterica* serovar Typhimurium, primers were designed for the *invA* gene. The sequences of primers were as follows: vB_Sen-TO17 Forward, 5′AGCGTTAGTTCTGTCCACCC3′; vB_Sen-TO17 Reverse, 5′CGCTGGCACTAATTTCGGTG3′; vB_SenM-2 Forward, 5′GCGCGACTTGTAAGATGCTG3′; vB_SenM-2 Reverse, 5′CCAATCAAGGGGCTTCTCGT3′; *S*. Typhimurium Forward, 5′AGCGTACTGGAAAGGGAAAG3′; *S*. Typhimurium Reverse, 5′ATACCGCCAATAAAGTTCACAAAG3′.

### Identification of phage DNA

The PCR reactions were performed to detect the DNA of bacteriophages vB_SenM-2 and vB_Sen-TO17 in the plasma of chickens. For this purpose, 10 μl of Color Taq PCR Master Mix (2×), 2 μl of specific forward and reverse primers each (final concentration of 1 μM), 5 μl of nuclease-free water, and 1 μl of matrix DNA (concentration 10 ng/μl)

were mixed. The reaction was conducted in Mastercycler® nexus (Eppendorf, Germany) with the following parameters: number of cycles−30; denaturation−15 s, 94 °C; annealing−15 s, 55 °C; extension −60 s, 72 °C. Then, the samples were loaded into wells of 1.5% agarose gel (agarose solution in Tris-octane-EDTA buffer (Bioshop, Canada) supplemented with SimplySafe™ (EURx, #E4600-02). Electrophoresis was set at 100 V for 30 min. DNA was visualized by UV light using a gel documentation system (FastGene FAS-DIGI PRO, Nippon Genetics Europe, Germany) with the following parameters: aperture 5.6 AV, exposure 1/40 TV, ISO 800.

### Transfection of chicken lymphoblasts and mouse splenocytes with bacterial and phage DNAs

Lymphoblasts and splenocytes ($3 \times 10^7$ cells per 15-cm plate for RNA isolation and dot-blot, or $1 \times 10^6$ cells per a well of 6- or 12-well plate for ELISA and microscopic studies) were cultured as described above. The medium was exchanged with a new one without serum and antibiotics, and the cells were transfected with DNA of bacteriophages vB_SenM-2 and vB_Sen-TO17 (the amount corresponding to $1 \times 10^{10}$ DNA molecules of each phage per $1 \times 10^7$ cells) or the corresponding amount of chromosomal DNA of *S. enterica* serovar Typhimurium strain 13, using TurboFect (Thermo Fisher Scientific, #R0531), according to the manufacturer's instructions. The efficiency of transfection with phage and bacterial DNAs was confirmed by PCR, using specific primers, described in the subsection 'Primer design'. Transfected cells were cultured for 24 h without or with the RNA polymerase III inhibitor CAS 577784-91-9 (Sigma-Aldrich, #557403) at a final concentration of 25 μM (which is close to the $IC_{50}$ value for human RNA polymerase III, equal to 27 μM).

### Reverse transcription-quantitative polymerase chain reaction (RT-qPCR)

The transfected (or non-transfected) cells, cultured as described above for 24 h, were collected, centrifuged at 4 °C, for 5 min, at $500 \times g$, and the tubes were immediately put on ice. Pellets were suspended in 0.5 ml of ice-cold TRIzol™ Reagent (Thermo Fisher Scientific, #15596026). After thawing, the mixture was homogenized with The Bead Ruptor Elite (Omni International, #SKU19-040E) using 4 beads per tube. Following centrifugation in a microfuge (maximal speed) for 5 min, supernatants were transferred to new tubes, 0.2 ml of cold chloroform was added, and after 5 min incubation, the samples were centrifuged as above. Upper fractions, containing RNA, were transferred to new tubes, 0.2 ml of 70% ethanol was added, put onto the column of the RNeasy Mini Kit (Qiagen, #74104), and purified according to the manufacturer's instruction. RT-qPCR analyses were performed using Light Cycler 480−II (Roche), with Sybr Mix (Roche) and following primers: GAPDH (1) Forward, 5′GTCTGGAGAAACCAGCCAAGT3′; GAPDH (1) Reverse, 5′CTCCTTGGATGCCATGTGGA3′; GAPDH (2) Forward, 5′CGTCTGGAGAAACCAGCCAA3′; GAPDH (2) Reverse, 5′TCCTTGGATGCCATGTGGAC3′; STING (1) Forward, 5′CAGCTGCATCTCACTGCTGC3′; STING (1) Reverse, 5′CAGGTGGAAGATCTCCTCGG3′; STINTG (2) Forward, 5′CAGCTGCATCTCACTGCTG3′; STING (2) Reverse, 5′AGGTGGAAGATCTCCTCGGC3′; IRF3 (1) Forward, 5′AAGGTCTTCTGGGCCTTGTC3′; IRF3 (1) Reverse, 5′AACTGGGGAACCAGCTTTACC3′; IRF3 (2) Forward, 5′AGAAGTGCAAGGTCTTCTGGG3′; IRF3 (2) Reverse, 5′AACTGGGGAACCAGCTTTAC3′; cGAS Forward, 5′GCTCCCAAGAGGAAACACC3′; cGAS Reverse, 5′GCTGCCAGTCTGTGTCACTT3′. Transcript abundance was estimated employing the $2^{-\Delta\Delta C(T)}$ method, using *GAPDH* as a reference gene.

### Detection and quantification of dsRNA in chicken and mouse cells

The abundance of dsRNA in chicken lymphoblasts and mouse splenocytes was estimated using fluorescent microscopy and dot-blot experiments. The cells were cultured and transfected (or not

transfected in control experiments) with phage or bacterial DNAs as described above. In some experiments, to minimize the background signal, $10^7$ cells were suspended in 0.5 ml of the Red blood cell (RBC) lysis buffer (eBioscience, #00-4300-54). Following 10 min incubation at room temperature, the samples were centrifuged for 5 min in a microfuge, and the pellet was suspended in the medium.

For fluorescent microscopy, cells were fixed with 4% paraformaldehyde in PBS (Thermo Fisher Scientific, #28906), rinsed with PBS and then with 0.1% Triton X-100 (Sigma, #X100-1GA), incubated in 0.5% BSA (Sigma-Aldrich, #A9418) for 1 h, and then incubated with anti-dsRNA clone rJ2 antibody (Sigma-Aldrich, #MABE1134) at the 1:100 ratio for 2 h. For detection of the rJ2 monoclonal antibody-specific signals, the samples were washed 4 times with 0.5% BSA and incubated for 1 h at room temperature with fluorophore-conjugated secondary Goat Anti-Mouse IgG (H + L), Highly Cross-Adsorbed antibody (Biotium, #20231-1) (1:500 ratio). To visualize cell nuclei, samples were incubated with the DAPI fluorescence dye (Invitrogen, #P36935). Coverslips were washed 4 times with PBS and adhered to glass slides with a mounting medium, and the next day, they were observed under the fluorescent microscope Leica DM4000 B.

The dot-blot experiments were performed using the above-described cells and antibodies, according to the previously described protocol[54]. Briefly, dilutions of RNA samples, purified as for RT-qPCR (see preceding subsection), were spotted (4 µl-drops) onto the nylon membrane and dried. Blocking non-specific sites on the membrane was conducted overnight using the blocking solution (50 µg/ml Sheared Salmon DNA (Thermo Fisher Scientific, # AM9680) in 1 x PBS-Tween (Thermo Fisher Scientific, #J60304.K3), 5% low-fat milk (Roth, # T145.1) w/v). Incubation with the primary antibody (1:1000) was conducted at room temperature for 2 h, the membrane was washed with PBS-Tween, and then incubated with the secondary antibody (1:10,000 in PBS-Tween, 2% low-fat dry milk w/v) at room temperature for 1 h. Following washing 3 times with PBS-Tween, the signals were detected using an Rtg membrane.

### Quantification and statistical analysis
The quantitative results are presented as mean values ± standard deviation (SD). For statistical analysis of the results, the SPSS 21.0 software (SPSS Inc., Armonk, NY, USA) was used. The normality of the distribution of variables was checked with the Kolmogorov–Smirnov test and the homogeneity of the variances with Levene's test. Once both assumptions were met, the analysis was carried out using ANOVA and *post hoc* Tukey's test. Otherwise (one or both assumption(s) was/were not met), the Kruskal–Wallis test and *post hoc* Dunn test were applied. Statistically significant differences were considered when $p < 0.05$.

### Reporting summary
Further information on research design is available in the Nature Portfolio Reporting Summary linked to this article.

## Data availability
Data generated in this study are included in the Source Data file provided with this paper. Source data are provided with this paper.

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

## Acknowledgements

The authors thank Gracja Topka-Bielecka for assistance in some experiments and Joanna Bart for professional corrections in English usage. This work was supported by the National Science Center (Poland) project grant no. 2017/27/B/NZ9/00393 to A.W.

## Author contributions

M.Po. participated in designing and performing experiments and in analyzing their results, as well as in reviewing and editing the manuscript; L.Ga. participated in designing and performing experiments and in analyzing their results, in preparing figures, as well as in reviewing and editing the manuscript; Ł.Gr. participated in designing and performing experiments and in analyzing their results, prepared drafts of figures, and participated in reviewing and editing the manuscript; J.M. participated in designing and performing experiments and in analyzing their results, as well as in reviewing and editing the manuscript; M.G. assisted in experimental procedures and was involved in data analysis; M.Pi. participated in discussions as well as in reviewing and editing the manuscript; K.P. was involved in data analysis, as well as in reviewing and editing the manuscript; G.W. invented the hypothesis on the mechanism of inhibition of the cGAS-STING pathway, participated in data analysis, and in drafting, reviewing and editing the manuscript; A.W. conceptualized the work, led and supervised the project, analyzed the data, participated in drafting the manuscript and prepared its final version.

## Competing interests

The authors declare no competing interests.
