## [Peer Review File · Nature Communications]

Bacteriophage DNA induces an interrupted immune response during phage therapy in a chicken modelREVIEWER COMMENTS

Reviewer #1 (Remarks to the Author):

This study is an important study that will clarify the relationship between phage and animals and determine the future direction of phage therapy. I have three comments.

1. It would be interesting to know how fecal microbiota and metabolites can change. Please do the shot-gun metagenomics by short-read sequencer and do the pathway analysis. Or please do 16S rRNA amplicon sequencing using long-read sequencing and predict the pathways.
2. Safety and efficacy of phages as a feed additive has been approved, and Proteon Pharmaceutical LTD. (<https://www.proteonpharma.com/>) has produced the food additive for avian Salmonella infections. According to your results, phage is not safe and would require a more detailed study in order to be marketable. In the end of your discussion, you need to discuss the current regulation for phage products and safety of phages. Probably, phages therapy should be divided into two types, as a drug and as a supplement. You can also discuss the phage therapy from this aspect.
3. Uniformity of words and characters are not consistent. Please fix them.

Reviewer #2 (Remarks to the Author):

The manuscript by Grabowski et al. describes the cell-mediated immune response to phage administration in a poultry model. Salmonella phages were orally administered to chickens with and without pathogen challenge and serum markers were assessed at intervals during and after treatment. Consistent with previous reports, phage alone did not induce inflammation. Phage did produce increased IFN-beta levels, which was associated with the TLR-9 pathway and not the cGAS pathway. Interaction of the phage with TLR-9 has been observed previously. This is an interesting study that specifically examines host cytokine responses to phage treatment, and goes a bit deeper than simply looking at major inflammatory cytokines.

The manuscript contains spelling and grammatical errors throughout the text, and should be closely proofread.

L106-108: This PCR appears to show presence/absence only, without quantification. Is it known what the detection limit of this PCR assay is? Was there any endogenous Salmonella in the phage-only treatment group that could have supported phage replication after the treatment ended? Detection of phage for so long after treatment seems unusual, as others have reported that phage tend not to persist in vivo without their host bacteria present.

L 104: the sampling days listed are different from those in the Methods and figures.

L120: IFN-beta can be produced by many cell types, is it possible to tell what cells produced IFN-beta observed here? Could the IFN-beta observed be produced by cells in the gut epithelium?

L245-270: The reasoning for why the phage DNA does not lead to IFN-beta production here is plausible, but the section could be condensed considerably, removing for example the review of basic transcriptional requirements at lines 255-259.

L330-334: Please provide the accession numbers for the genome sequences of the phages used in the study.

L361-363: what was the endotoxin level (or range of levels) in the final administered phage mixture?

L 338: was the *S. Typhimurium* strain used for poultry challenge sensitive to both phages used?

L 386-388: it is not clear here how this procedure ensures "maximum disintegration of blood cells", would detergent or freeze-thaw cycles not provide complete lysis?

Reviewer #3 (Remarks to the Author):

This study uses chickens infected with *Salmonella enterica* to examine the immune response to bacteriophages. The authors report that phages are generally anti-inflammatory and dampen the immune response to a bacterial pathogen. Mechanistically, they report that STING-mediated signal transduction was blocked at the stage of IRF3 phosphorylation and that TLR9 signaling was stimulated, resulting in elevated interferon β and anti-inflammatory effects.

This is a fascinating study with direct relevance to an important area of experimental medicine - phage therapy. While some reports suggest that phages are immune activating (including work with phage vaccines), phage therapy is generally safe and well-tolerated. The dampening of anti-inflammatory effects associated with bacterial pathogens described here may be why.

This study has several strengths. These include the outstanding model system and robust experimental design, the focus on innate immune pathways and signaling responses, and the strong results. The model that DNA phages are triggering TLR9 mediated immunity that antagonizes bacterial responses is compelling. The data are clearly presented and the work is well-written.

However, these results are mostly observational. Without functional interventions it is difficult to understand how these effects work. For example, the authors attribute these effects to phage therapy but it is possible that these effects are mediated by shifts in the microbiota or immune polarization. These options are possible particularly given that the responses persist long after the end of phage therapy. It is also uncertain what cell types are responsible or how these effects come together; simultaneous effects on TLR9 and cGAS/STING can be inferred but some parts of these pathways are unclear. Ultimately, these findings are potentially important but challenging to interpret.

Major comments:

1. One of the key findings of this work are phage effects on cGAS/STING. The investigators conclude that cGAS/STING is triggered but cGAS and STING levels are low, perhaps consistent with the downregulation. Moreover, IRF3 levels are low, suggesting that this pathway is blocked. Can the investigators demonstrate more directly (i.e. in a cell reporter system or in chicken cells *ex vivo*) that these phages trigger cGAS/STING?

2. The investigators postulate that these phages are recognized by RNA polymerase III and that this blocks IRF3 signaling. This is an intriguing possibility but it is not shown here. Can the investigators demonstrate this? Perhaps in the context of the proposed cellular studies?

3. It would be important to attribute these effects to one phage or to conclude that both phages have these effects. A cocktail of phages is used in this study. Can the investigators show effects for both phages independently?

4. It is unclear what cell types are affected here or whether these are immune cells. Can the investigators show that these pathways are relevant to monocytes or another immune cell population?

5. How do the authors understand the lasting effects on immune polarization in these animals? Why are short-acting signaling pathway components (e.g. phospho-NFkB) altered a month after phage and bacterial treatment? This needs to be discussed.

Reviewer #4 (Remarks to the Author):

The authors have presented a manuscript entitled "Bacteriophage DNA recognition by an animal immune system during phage therapy in the chicken model". The authors carried out a single large experiment with 100 chicks infected with salmonella, bacteriophage, or salmonella + bacteriophage, or PBS controls. The authors then took blood samples at several time points and measures various parameters by ELISA. The authors concluded by quantifying various innate signalling proteins, that there are different pathways triggered by the bacteria and bacteriophages, and that the predominant pathways are those activated by phage DNA. The premise of this manuscript has merit. There are safety implications of phage therapy if these viruses can trigger pattern recognition receptors, leading to generalised inflammation and/or interferon production during treatment. The chicken, salmonella infection model used here is a relevant and interesting model to analyse this question. There are, however, some major issues in the experimental design and execution that require addressing before the manuscript can be considered acceptable for publication.

1) The data consists primarily of non-standard ELISA kits for the quantification of intracellular signalling molecules. There are no details given for the product numbers or provenance of these kits. Given their absolute reliance on these kits for the validity of the conclusions of the manuscript, it is essential that the authors provide full details of each individual kit used (including source, catalogue numbers, etc.) and validation of the specificity of the antibodies contained in each kit proving they are reliable to measure and quantify the target proteins as indicated.

2) There is no data normalisation in the ELISA experiments to indicate how many cells were present in the blood at each time point with each treatment. Without knowing the relative numbers of cells in each condition it is impossible to interpret the relative activation of signalling pathways between treatments. These data should be presented 'per cell' or 'per 10^6 cells' or with some protein-based normalisation in order to compare the relative responses to individual treatments.

3) There is relevant information provided by quantifying IFN α /b and other secreted proteins and (notwithstanding points 1 and 2) the quantification of pIRF3/7 and pNF-kb. However, quantifying the levels of other signalling proteins - receptors like TLR9, cGAS etc, adaptors proteins like STING, DDX41 etc. - does not inform on the activation of any specific signalling pathway and, hence cannot be concluded as such.

Because of these major issues, the data presented does not support the conclusions and I cannot recommend the manuscript for publication in its current format.

NCOMMS-23-22231: Response to reviews

Reviewer #1 (Remarks to the Author):

This study is an important study that will clarify the relationship between phage and animals and determine the future direction of phage therapy. I have three comments.

1. It would be interesting to know how fecal microbiota and metabolites can change. Please do the shot-gun metagenomics by short-read sequencer and do the pathway analysis. Or please do 16S rRNA amplicon sequencing using long-read sequencing and predict the pathways.

RESPONSE:

We fully agree with the reviewer that the proposed experiments are important. In fact, investigation of the microbiomes, based on 16S rRNA analyses, were performed by our team, and the results have been published recently (ref. 29 in the manuscript). In the revised manuscript, we have mentioned these results, and referred readers to the previously published article (ref. 29). The following text has been included into the manuscript (lines: 77-81):

“Analyses of chicken gut microbiomes demonstrated that treatment with the phage cocktail caused only temporary changes in the composition of the microbiota which normalized within a couple of weeks, with predominance of Lactobacillaceae and Enterobacteriaceae. This was in contrast to birds treated with antibiotics (enrofloxacin or colistin) where the microbiomes remained significantly changed until the end of the experiment.²⁹”

2. Safety and efficacy of phages as a feed additive has been approved, and Proteon Pharmaceutical LTD. (<https://www.proteonpharma.com/>) has produced the food additive for avian Salmonella infections. According to your results, phage is not safe and would require a more detailed study in order to be marketable. In the end of your discussion, you need to discuss the current regulation for phage products and safety of phages. Probably, phages therapy should be divided into two types, as a drug and as a supplement. You can also discuss the phage therapy from this aspect.

RESPONSE:

We thank the reviewer for this comment. However, we cannot agree that according to our results “phage is not safe”. In fact, the safety of the phage cocktail (used in this work) for chickens has been demonstrated previously (ref. 30 in the manuscript). As noted by another reviewer (Reviewer #3) and in agreement with interpretation of the presented

results by the authors of this paper, what is actually demonstrated in this manuscript is the molecular mechanism by which the administration of bacteriophages is safe and does not result in a strong anti-viral (or anti-microbial) immune response of animals.

As requested by the reviewer, at the end of the Discussion of the revised manuscript, we present and discuss the current regulations for phage products and safety of phages in the light of phage therapy and prevention against bacterial infections. In fact, recent regulations of European Union and European Medicines Agency defined the requirements for both these types of potential applications. The approval of bacteriophages as feed supplements (additives) is possible under Regulation (EC) No. 1831/2003 of the European Parliament and of the Council of September 22, 2003 on additives for use in animal nutrition. The major requirements for such a use of bacteriophages are as follows: (i) the phage product must be covered by an authorization issued in accordance with the above regulation; (ii) the conditions for the use, specified in the above regulation, must be fulfilled; and (iii) the labeling conditions, specified in the above regulation, must be complied with. Moreover, during the legislative process, a manufacturer must demonstrate that the additive is effective in eradicating the pathogen and is safe for the animal body. Such an additive must comply with the requirements of Article 5(2) of Regulation (EC) No. 1831/2003, i.e. (i) it must not have harmful effects on animal health, human health or the environment; (ii) it must not be presented in a manner that may mislead the user; (iii) it must not be harmful to the consumer; and (iv) it must have the attribute specified in paragraph 3 of this article. It is worth noting that the approval of any feed supplement (additive) depends on the evaluation of every product by European Food Safety Authority Panel on Additives and Products or Substances used in Animal Feed. On the other hand, on October 23, 2023, the European Medicines Agency's Regulation EMA/CVMP/NTWP/32862/2022 came into force on guidelines for the quality, safety and efficacy of veterinary medicinal products specifically for phage therapy. The requirements are significantly more strict and difficult to fulfill than those for feed supplements. Bacteriophages used as therapeutic preparations must be manufactured in accordance with Good Manufacturing Practice. In reality, the manufacturing process is very complex, as it involves the completion of expensive tasks, including training and validation of personnel, monitoring of appropriate apparatus, validation of quality control procedures and methods, and inspection by the relevant authorities. Furthermore, according to the regulation, the phage preparation must undergo an evaluation of its efficacy and safety as part of preclinical studies on the target animal species. This stage aims to determine pharmacokinetics, target animal safety, the dose of the preparation used, and the risk of bacterial resistance to the phage preparation. The legislator also allows preclinical studies to be conducted on non-target organisms, or validated in vitro models, to demonstrate the mechanism of action of the preparation.

Since this paper is focused on molecular mechanisms of the immune response of animal cells to bacteriophages, rather than on phage therapy itself or on formal regulations, in

the revised manuscript we present a shortened version of the above explanation. The newly introduced paragraph reads as follows (lines 414-430):

“Demonstration of the molecular mechanism by which bacteriophage DNA does not induce a strong antimicrobial immune response through the cGAS-STING pathway, together with the documented production of anti-inflammatory cytokines in the presence of bacteriophage DNA, can explain the safety of administration of phages to animals, described previously.³⁰ This can be important in light of introduction of the phage therapy into veterinary medicine. It is worth to mention that bacteriophages may be considered for their use as either feed supplements (additives) or therapeutics. Recent regulations of the European Union and European Medicines Agency defined the requirements for both these types of potential applications. The approval of bacteriophages as feed supplements (additives) is possible under Regulation (EC) No. 1831/2003 of the European Parliament and of the Council of September 22, 2003, on additives for use in animal nutrition. Nevertheless, on October 23, 2023, the European Medicines Agency's Regulation EMA/CVMP/NTWP/32862/2022 came into force on guidelines for the quality, safety and efficacy of veterinary medicinal products specifically for phage therapy. The requirements are significantly stricter and more difficult to fulfill than those for feed supplements, including safety issues. Therefore, studies like the work described in this report are important in the way that may lead to approval of the use of bacteriophages in the prevention or treatment of animals against bacterial infections.”

3. Uniformity of words and characters are not consistent. Please fix them.

RESPONSE:

We have followed the reviewer's recommendation, and the uniformity of the words and characters was checked and corrected. In fact, the English usage has also been corrected by a person professional in this language.

Reviewer #2 (Remarks to the Author):

The manuscript by Grabowski et al. describes the cell-mediated immune response to phage administration in a poultry model. Salmonella phages were orally administered to chickens with and without pathogen challenge and serum markers were assessed at intervals during and after treatment. Consistent with previous reports, phage alone did not induce inflammation. Phage did produce increased IFN-beta levels, which was associated with the TLR-9 pathway and not the cGAS pathway. Interaction of the phage with TLR-9 has been observed previously. This is an interesting study that specifically examines host cytokine responses to phage treatment, and goes a bit deeper than simply looking at major inflammatory cytokines.

The manuscript contains spelling and grammatical errors throughout the text, and should be closely proofread.

RESPONSE:

The manuscript has been corrected for spelling and grammatical errors by a person proficient with English.

L106-108: This PCR appears to show presence/absence only, without quantification. Is it known what the detection limit of this PCR assay is? Was there any endogenous *Salmonella* in the phage-only treatment group that could have supported phage replication after the treatment ended? Detection of phage for so long after treatment seems unusual, as others have reported that phage tend not to persist in vivo without their host bacteria present.

RESPONSE:

We thank the reviewer for this comment. Indeed, we have tested the detection limit of the PCR assay. The results are described now in the manuscript (the detection limits in the PCR analyses were determined to be 100 fg and 10 fg for vB_SenM-2 and vB_Sen-TO17 phage DNAs, respectively (corresponding to 6.19×10^2 and 2.08×10^2 molecules of genomic DNA of vB_SenM-2 and vB_Sen-TO17, respectively; lines 111-114) and presented in Supplementary Material (Supplementary Figure S1).

As described in the previously published article (ref. 29 in this manuscript), no *Salmonella* cells could be detected in the phage-only treatment group. Therefore, the presence of phage DNA in chicken blood for over a month cannot be explained by phage multiplication in the body of birds. However, we would like to stress that we have tested the presence of bacteriophages in chicken feces and stomach, and in the phage-only treatment group, the bacteriophages were present for only 1-2 weeks after their administration (ref. 29). On the other hand, it is important to remind that in this manuscript, we did not monitor the presence of bacteriophages in blood, but rather the presence of phage DNA. We agree that these results were quite surprising, nevertheless, it is worth noting that as demonstrated previously (ref. 29), following oral administration, bacteriophages penetrated various organs of chickens quite effectively. Therefore, the presented results indicate that phage genetic material can persist in the blood for a relatively (perhaps surprisingly) long time.

L 104: the sampling days listed are different from those in the Methods and figures.

RESPONSE:

We are sorry, this was a typographical error in the text. It was corrected now, and the sampling days mentioned in the text (line 109) correspond to those in the Methods and figures.

L120: IFN-beta can be produced by many cell types, is it possible to tell what cells produced IFN-beta observed here? Could the IFN-beta observed be produced by cells in the gut epithelium?

RESPONSE:

IFN α and IFN β are produced by a wide range of cells such as macrophages, fibroblasts, endothelial cells and plasmacytoid dendritic cells. Hypothetically, all these cells are equipped with the molecular machinery to recognize viral infection and express type I IFN (IFN α and IFN β) and type II IFN in response. Due to the fact that class I IFNs can be secreted by various types of cells, based on the results presented in our manuscript, it is difficult to clearly state which cell type(s) had the greatest share in the release of IFN β , but it cannot be excluded that intestinal epithelial cells play a dominant role. Evidently, further extensive studies would be necessary to answer this question.

L245-270: The reasoning for why the phage DNA does not lead to IFN-beta production here is plausible, but the section could be condensed considerably, removing for example the review of basic transcriptional requirements at lines 255-259.

RESPONSE:

In response to Reviewer #3, we have performed additional experiments which confirmed our hypothesis (just to note the typographical error in the reviewer's comment, as we demonstrated that phage DNA did lead to production of IFN-beta, but not IFN-alpha). Thus, the text and the part of Discussion mentioned by the Reviewer #2 have been modified to reflect the updated state of our knowledge. Nevertheless, in our opinion, a short description of the regulatory system is required, especially for those interested in our paper but not being experts in immunology (like some phage biologists or geneticists).

L330-334: Please provide the accession numbers for the genome sequences of the phages used in the study.

RESPONSE:

The GenBank accession numbers for genomes of the studied phages (KX171211 and MT012729 for vB_SenM-2 and vB_Sen-TO17, respectively) are provided (lines 482 and 484), as requested by the reviewer.

L361-363: what was the endotoxin level (or range of levels) in the final administered phage mixture?

RESPONSE:

According to the reviewer's request, we included the measured levels of the endotoxin. We have described these results in lines 515-517 as follows: "The levels were 0.156 ± 0.062 and 0.085 ± 0.043 EU/kg/h, for vB_SenM-2 and vB_Sen-TO17, respectively (with the negative control <0.05 EU/kg/h, and the upper acceptable limit of the enterotoxin concentration equal to 5.0 EU/kg/h)."

L 338: was the *S. Typhimurium* strain used for poultry challenge sensitive to both phages used?

RESPONSE:

Yes, this strain was sensitive to both phages. This is now indicated in the revised manuscript (lines 490-491: "this strain is sensitive to both vB_SenM-2 and vB_Sen_TO17, as reported previously.²⁹").

L 386-388: it is not clear here how this procedure ensures "maximum disintegration of blood cells", would detergent or freeze-thaw cycles not provide complete lysis?

RESPONSE:

Obviously, the use of a strong detergent would allow the complete lysis of cells, however, we were afraid that the presence of a detergent might influence the results of measurements of the levels of specific proteins in ELISA. Similarly, although to a lesser extent, this might apply to freeze-thaw cycles. Therefore, we decided to achieve the disruption of the cell membranes and liberation of proteins by choosing appropriate centrifugation conditions. Indeed, properly selected centrifugation parameters and appropriate tubes did guarantee both effective disintegration of cells and the proper separation of blood components. In this way, we selected optimal conditions for determination of the studied parameters.

Reviewer #3 (Remarks to the Author):

This study uses chickens infected with *Salmonella enterica* to examine the immune response to bacteriophages. The authors report that phages are generally anti-inflammatory and dampen the immune response to a bacterial pathogen. Mechanistically, they report that STING-mediated signal transduction was blocked at the stage of IRF3 phosphorylation and that TLR9 signaling was stimulated, resulting in elevated interferon β and anti-inflammatory effects.

This is a fascinating study with direct relevance to an important area of experimental medicine - phage therapy. While some reports suggest that phages are immune activating (including work with phage vaccines), phage therapy is generally safe and well-tolerated. The dampening of anti-inflammatory effects associated with bacterial pathogens described here may be why.

This study has several strengths. These include the outstanding model system and robust experimental design, the focus on innate immune pathways and signaling responses, and the strong results. The model that DNA phages are triggering TLR9 mediated immunity that antagonizes bacterial responses is compelling. The data are clearly presented and the work is well-written.

However, these results are mostly observational. Without functional interventions it is difficult to understand how these effects work. For example, the authors attribute these effects to phage therapy but it is possible that these effects are mediated by shifts in the microbiota or immune polarization. These options are possible particularly given that the responses persist long after the end of phage therapy. It is also uncertain what cell types are responsible or how these effects come together; simultaneous effects on TLR9 and cGAS/STING can be inferred but some parts of these pathways are unclear. Ultimately, these findings are potentially important but challenging to interpret.

Major comments:

1. One of the key findings of this work are phage effects on cGAS/STING. The investigators conclude that cGAS/STING is triggered but cGAS and STING levels are low, perhaps consistent with their downregulation. Moreover, pIRF3 levels are low, suggesting that this pathway is blocked. Can the investigators demonstrate more directly (i.e. in a cell reporter system or in chicken cells ex vivo) that these phages trigger cGAS/STING?

RESPONSE:

As suggested by the reviewer, we have performed additional experiments using the DT40 chicken lymphoblast line. Moreover, we confirmed (using mouse splenocytes) that the mechanism operates also in mammalian cells. This is described in following fragments of the revised manuscript:

Lines 200-211: "To confirm that the response of cultured chicken lymphoblasts to phage and bacterial DNAs was comparable to that observed in the blood samples of birds infected with *S. enterica* and/or treated with the phage cocktail, we have determined the efficiency of expression of genes coding for cGAS, STING, and IRF3 using RT-qPCR. The obtained results confirmed that the tendencies of changes in the expression of tested genes determined in the transfected lymphoblasts were comparable to those observed in samples of chicken blood when levels of corresponding gene products were estimated (Fig. 4B). Moreover, we have measured levels of the phosphorylated form of IRF3 in the transfected lymphoblasts, to show that they are significantly elevated in the presence of bacterial DNA, but not in the presence of phage DNA (Fig. 4C). These results indicated that the immune response to genomic DNA of *S. enterica* and to phage DNA observed in the transfected lymphoblasts is comparable to that observed in chickens."

Lines 239-248: "We confirmed that similar to chickens, mouse splenocytes responded to the presence of bacterial DNA by enhanced production of both IFN α and IFN β , while only IFN β (but not IFN α) was produced after transfection of these cells with bacteriophage DNA (Supplementary Fig. S4). Since in the case of mouse splenocytes the transfection efficiencies by bacterial and phage DNAs were similar to those observed for chicken lymphoblasts, and modulations of levels of selected proteins involved in the immune response were comparable between mouse and chicken cells (Supplementary Fig. S5), we concluded that responses of mammalian and bird cells to bacterial and phage DNAs are comparable, including the inhibition of the cGAS-STING signal transduction pathway at the stage of IRF3 phosphorylation in the presence of bacteriophage DNA."

2.The investigators postulate that these phages are recognized by RNA polymerase III and that this blocks IRF3 signaling. This is an intriguing possibility but it is not shown here. Can the investigators demonstrate this? Perhaps in the context of the proposed cellular studies?

RESPONSE:

This is a very important point, and we have followed the reviewer's recommendation to demonstrate that the proposed hypothesis is plausible. Therefore, we have performed a series of additional experiments. Indeed, the obtained results indicated that the tested hypothesis is correct, demonstrating that the molecular mechanism of the interruption of the cGAS/STING signal transduction at the IRF3 phosphorylation stage consists of inability of RNA polymerase III to recognize phage DNA and to produce dsRNA. These experiments are described in the revised manuscript and shown in new figures and table (Figs. 4, 5, S4, S5, S6, Table S1). The new text is included in lines 173-255 and reads as follows:

"The mechanism of blocking the IRF3 phosphorylation in response to bacteriophage DNA presence

Since the inhibition of IRF3 phosphorylation appeared to be crucial in blocking the immune response of chicken cells to the presence of bacteriophage DNA, we aimed to investigate the molecular mechanism of this regulatory process. Earlier studies demonstrated that RNA polymerase III is able to transcribe cytosolic DNA, thus synthesizing 5'-triphosphate double-stranded RNA (dsRNA).^{31,32} Such dsRNA, after being recognized by RIG-I-like receptors (RLRs), creates a signal stimulating the adaptor proteins (MAVS) which together with RLRs contribute to the activation of TBK1 and IKKε which is necessary to IRF3 phosphorylation.³³ We hypothesized that bacteriophage DNA cannot be effectively transcribed by RNA polymerase III, as no sequences resembling eukaryotic promoters and specific AT-rich regions are present in their relatively short genomes (in contrast, eukaryotic viruses contain specific promoters, and bacterial genomes are large enough to contain random sequences resembling RNA polymerase III-specific promoters). If this hypothesis is true, in the response to bacteriophage DNA, RLRs-MAVS-mediated activation of TBK1 and IKKε might be ineffective, resulting in impaired IRF3 phosphorylation and its inability to translocate to the nucleus.

To test the above hypothesis, we have used a line of chicken lymphoblasts (the DT40 line), transfected with either bacteriophage DNA (isolated from both vB_SenM-2 and vB_Sen-TO17) or genomic DNA from the *S. enterica* strain (non-transfected cells were employed as a negative control). We demonstrated that the efficiency of transfection by both phage genomes and the fragmented bacterial genome was high enough to indicate the efficient entrance of foreign DNAs into chicken cells (Fig. 4A). Moreover, the absence of considerable amounts of the endotoxin in all tested DNA samples was confirmed in the LAL test. The determined values were 0.87, 0.98, and 0.96 EU/mL for DNAs of vB_SenM-2, vB_Sen-TO17, and *S. enterica*, respectively, where values <1 EU/mL are considered as ultra-pure samples.³⁴

To confirm that the response of cultured chicken lymphoblasts to phage and bacterial DNAs was comparable to that observed in the blood samples of birds infected with *S. enterica* and/or treated with the phage cocktail, we have determined the efficiency of expression of genes coding for cGAS, STING, and IRF3 using RT-qPCR. The obtained results confirmed that the tendencies of changes in the expression of tested genes (or lack of changes) determined in the transfected lymphoblasts were comparable to those observed in samples of chicken blood when levels of corresponding gene products were estimated (Fig. 4B). Moreover, we have measured levels of the phosphorylated form of IRF3 in the transfected lymphoblasts, to show that they are significantly elevated in the presence of bacterial DNA, but not in the presence of phage DNA (Fig. 4C). These results indicated that the immune response to genomic DNA of *S. enterica* and to phage DNA observed in the transfected lymphoblasts is comparable to that observed in chickens.

To test whether dsRNA can be synthesized by RNA polymerase III in the presence of either phage or bacterial DNA in chicken cells, we have used an antibody specific to dsRNA. The detection of specific signals was performed using fluorescent microscopy

and observation of transfected cells, and the dot-blot procedure with the material obtained after cell lysis. When cells transfected with fragments of the *S. enterica* chromosome were investigated, intensive signals derived from dsRNA were observed in both types of experiments (fluorescent microscopy and dot-blot), contrary to non-transfected cells when the signals were at the background level (Fig. 5 A-D). When an inhibitor of the RNA polymerase III was used (at the final concentration corresponding to the IC50 value; this concentration was chosen to avoid a significant inhibition of the cell growth, otherwise observed in cultures incubated for the time of the whole experiment at higher concentrations), the intensities of dsRNA-specific signals were significantly (about 50%) lower, indicating that the detected dsRNA molecules were produced at least predominantly by this type of RNA polymerase (Fig. 5 A-D). Different results were obtained in experiments with chicken lymphoblasts transfected with phage DNA. In this case, no significant increase in the levels of dsRNA could be observed in both types of experiments, and the abundance of dsRNA in cells bearing phage DNA was statistically indistinguishable from that in control (non-transfected) cells (Fig. 5 A-D). These results corroborated the proposed hypothesis that the block in the signal transduction in response to the presence of phage DNA, observed at the stage of phosphorylation of IRF3, is due to the inability of production of dsRNA by RNA polymerase III under these conditions, likely due to a lack of recognizable sequences in bacteriophage genomes.

To test whether the mechanism of blocking the cGAS-STING pathway is specific to chickens or it is a more common phenomenon, we have repeated the kinds of experiments described in the above paragraph with mouse splenocytes transfected with either phage or bacterial DNA. Splenocytes were isolated from wild-type mice which were tested for their general health conditions to confirm the consistency of all parameters with the norms (Supplementary Table S1).

We confirmed that similar to chickens, mouse splenocytes responded to the presence of bacterial DNA by enhanced production of both IFN α and IFN β , while only IFN β (but not IFN α) was produced after transfection of these cells with bacteriophage DNA (Supplementary Fig. S4). Since in the case of mouse splenocytes the transfection efficiencies by bacterial and phage DNAs were similar to those observed for chicken lymphoblasts, and modulations of levels of selected proteins involved in the immune response were comparable between mouse and chicken cells (Supplementary Fig. S5), we concluded that responses of mammalian and bird cells to bacterial and phage DNAs are comparable, including the inhibition of the cGAS-STING signal transduction pathway at the stage of IRF3 phosphorylation in the presence of bacteriophage DNA. Therefore, we have tested the efficiency of dsRNA production by RNA polymerase III in response to the presence of bacterial and phage DNAs in mouse splenocytes. In these experiments, results similar to those obtained with chicken lymphoblasts were evident. Namely, dsRNA was efficiently produced in cells transfected with bacterial DNA, but not in those transfected with phage DNA (Supplementary Fig. S6). Therefore, we conclude that the

immune responses to phage DNA, including the mechanism of the inhibition of the cGAS-STING pathway, are similar in both chickens and mice..”

3. It would be important to attribute these effects to one phage or to conclude that both phages have these effects. A cocktail of phages is used in this study. Can the investigators show effects for both phages independently?

RESPONSE:

This is an interesting question of the reviewer. However, we have to consider two points. First, from the point of view of those interested in applications of phage therapy, the use of phage cocktails and determination of their effects is definitely more appropriate than evaluation of effects of single phages. This is because using of two or more phages in a cocktail significantly reduces the possibility of selection of phage-resistant mutants of bacteria which is crucial for a high efficacy of phage therapy. Therefore, we investigated the phage cocktail rather than two separated bacteriophages. Second, we assume that the proposed mechanism is common for most or all phages bearing dsDNA genomes, rather than specific for any unique bacteriophage. Indeed, the results of our experiments, especially those suggested by the reviewer and performed during revision, indicated that no dsRNA-specific signals could be detected after transfection of chicken cells with the mixture of genomes of two tested bacteriophages. If the response to one phage was different than that to another phage, an appearance of elevated amounts of dsRNA should be observed, which was not the case here. Therefore, we conclude that the described mechanism is common for the two tested phages, and possibly to most (if not all) other bacteriophages bearing dsDNA genomes.

4. It is unclear what cell types are affected here or whether these are immune cells. Can the investigators show that these pathways are relevant to monocytes or another immune cell population?

RESPONSE

This is another interesting question. Indeed, we have investigated the summary of effects observed in blood cells of chicken. However, during the revision, we have demonstrated that the investigated mechanism operates in specific type of cells - chicken lymphoblasts. Moreover, we also demonstrated that the mechanism occurs also in mouse splenocytes which are a mixture of different types of cells (T and B lymphocytes, dendritic cells and macrophages). Therefore, determination of specific types of cells where the pathways are relevant to remains to be determined in further studies.

5. How do the authors understand the lasting effects on immune polarization in these animals? Why are short-acting signaling pathway components (e.g. phospho-NFκB) altered a month after phage and bacterial treatment? This needs to be discussed.

RESPONSE:

The question about the short-acting signaling pathway components (e.g. phosphorylated NF-κB) which are altered a month after phage and bacterial treatment is obviously relevant. Although some of the tested factors are short-acting signaling pathway components (like phospho-NF-κB), one should note that in the described experiments, administration of bacteriophages was not strictly synchronized. In fact, the cocktail of bacteriophages was administered orally every day for two weeks (see Figure 1A). Then, phage DNA was detected in samples of blood even after 19 days after the last administration. Therefore, one should expect a continuous stimulation of the immune system during this period, rather than a synchronized short-term effect. In other words, alteration of short-acting signaling pathway components can be visible during a relatively long time if the administration of bacteriophages is not strictly synchronized and conducted as a single event, but rather extended in time. Under such conditions, the cells can be continuously stimulated by the phages (or their DNA) present in the blood, resulting in long-term alterations in levels/activities of short-acting signaling pathway components. In other words, the cells were stimulated again and again which resulted in several rounds of appearance of changes in short-acting signaling pathway components.

Reviewer #4 (Remarks to the Author):

The authors have presented a manuscript entitled "Bacteriophage DNA recognition by an animal immune system during phage therapy in the chicken model". The authors carried out a single large experiment with 100 chicks infected with salmonella, bacteriophage, or salmonella + bacteriophage, or PBS controls. The authors then took blood samples at several time points and measures various parameters by ELISA. The authors concluded by quantifying various innate signalling proteins, that there are different pathways triggered by the bacteria and bacteriophages, and that the predominant pathways are those activated by phage DNA. The premise of this manuscript has merit. There are safety implications of phage therapy if these viruses can trigger pattern recognition receptors, leading to generalised inflammation and/or interferon production during treatment. The chicken, salmonella infection model used here is a relevant and interesting model to analyse this question. There are, however, some major issues in the experimental design and execution that require addressing before the manuscript can be considered acceptable for publication.

1) The data consists primarily of non-standard ELISA kits for the quantification of

intracellular signalling molecules. There are no details given for the product numbers or provenance of these kits. Given they absolute reliance on these kits for the validity of the conclusions of the manuscript, it is essential that the authors provide full details of each individual kit used (including source, catalogue numbers, etc.) and validation of the specificity of the antibodies contained in each kit proving they are reliable to measure and quantify the target proteins as indicated.

RESPONSE:

We understand the reviewer's doubts. However, Shanghai Coon Koon Biotech Co., Ltd. was the only company which could supply us with tests specific for most of tested chicken proteins involved in the immune response (there are many companies providing such tests for mammalian proteins, but not chicken ones). In the revised manuscript, we provide catalogue numbers of all used kits (lines 561-581). We have asked this company to provide full details of each individual kit used and validation of the specificity of the antibodies contained there. However, the representative of the company answered that it is impossible due to the "company's trade confidential information". Nevertheless, the company provided a certificate for all their products which have been used in this work, ensuring that they are high quality assays, validated and confirmed to be reliable. This certificate is attached as a supplementary material "for review only, not for publication". We also provide results of control/preliminary experiments with the use of these kits where calibration curves for concentrations of tested proteins were obtained (a supplementary material "for review only, not for publication"). Moreover, we found in the literature over 90 published articles where the products of this manufacturer were used.

2) There is no data normalisation in the ELISA experiments to indicate how many cells were present in the blood at each time point with each treatment. Without knowing the relative numbers of cells in each condition it is impossible to interpret the relative activation of signalling pathways between treatments. These data should be presented 'per cell' or 'per 10^6 cells' or with some protein-based normalisation in order to compare the relative responses to individual treatments.

RESPONSE:

As requested by the reviewer, the results of the mentioned experiments are presented in the revised manuscript 'per 10^6 cells' (see all relevant Figures, i.e. Figs. 1, 2, 3, 6, 7, S2, S3). We agree that this kind of presentation shows the actual results in a clearer and more appropriate form. Nevertheless, since cell counts were similar in all the experimental variants, the final results do not differ considerably from those presented initially in the first version of the manuscript.

3) There is relevant information provided by quantifying IFN α /b and other secreted proteins and (notwithstanding points 1 and 2) the quantification of pIRF3/7 and pNF- κ b. However, quantifying the levels of other signalling proteins - receptors like TLR9, cGAS etc, adaptor proteins like STING, DDX41 etc. - does not inform on the activation of any specific signalling pathway and, hence cannot be concluded as such.

RESPONSE:

Obviously, we agree with this comment of the reviewer. Therefore, it is true that in the case of the signaling proteins, their modifications (like phosphorylation and others) are crucial rather than their absolute levels. This is why we have also tested these modifications using specific antibodies. We are aware that the descriptions included in the original manuscript might be unclear in this matter, therefore, the text has been carefully checked and corrected during revision to eliminate any uncertainties related to the described problem. Specifically, following explanations have been included:

Lines 153-155: "Nonetheless, it should be noted that not the specific levels of cGAS and STING, but their specific activities are crucial in the signal transduction process."

Lines 325-326: "However, it is the activity of cGAS (stimulated by DNA binding) rather than its level which is crucial for cGAMP synthesis."

REVIEWERS' COMMENTS

Reviewer #1 (Remarks to the Author):

The manuscript has been improved a lot.

I have no further comments.

Reviewer #2 (Remarks to the Author):

I would like to thank the authors for their revisions to the manuscript. I have only a few additional minor comments.

I suggest re-wording the title to "in a chicken model" instead of "in the chicken model". Related to this, there are still a number of minor grammar issues and awkward wording in the text (e.g., line 419, "worth to mention" should be "worth mentioning").

Reviewer #3 (Remarks to the Author):

The authors have addressed my comments.

Reviewer #4 (Remarks to the Author):

The authors have generated a revised manuscript that satisfactorily addresses the points raised in the initial review and so I recommend publication.

RESPONSE TO REVIEWS

Reviewer #1 (Remarks to the Author):

The manuscript has been improved a lot.
I have no further comments.

RESPONSE:

We are grateful to the reviewer for the positive opinion, and once again, we thank this reviewer for useful comments included in the previous review round. Since this reviewer accepted this paper as it is, we did not introduce further changes.

Reviewer #2 (Remarks to the Author):

I would like to thank the authors for their revisions to the manuscript.

RESPONSE:

We are grateful to the reviewer for the positive opinion, and once again, we thank this reviewer for useful comments included in the previous review round.

I have only a few additional minor comments.
I suggest re-wording the title to "in a chicken model" instead of "in the chicken model".

RESPONSE:

The title has been changed according to the reviewer's recommendation.

Related to this, there are still a number of minor grammar issues and awkward wording in the text (e.g., line 419, "worth to mention" should be "worth mentioning").

RESPONSE:

The minor English errors have been corrected according to the reviewer's recommendations.

Reviewer #3 (Remarks to the Author):

The authors have addressed my comments.

RESPONSE:

We are grateful to the reviewer for the positive opinion, and once again, we thank this reviewer for useful comments included in the previous review round.

Reviewer #4 (Remarks to the Author):

The authors have generated a revised manuscript that satisfactorily addresses the points raised in the initial review and so I recommend publication.

RESPONSE:

We are grateful to the reviewer for the positive opinion, and once again, we thank this reviewer for useful comments included in the previous review round.